# Fully Inductive Node Representation Learning via Graph View Transformation

## Abstract

Generalizing a pretrained model to unseen datasets without retraining is an essential step toward a foundation model. However, achieving such cross-dataset, fully inductive inference is difficult in graph-structured data where feature spaces vary widely in both dimensionality and semantics. Any transformation in the feature space can easily violate the inductive applicability to unseen datasets, strictly limiting the design space of a graph model. In this work, we introduce the *view space*, a novel representational axis in which arbitrary graphs can be naturally encoded in a unified manner. We then propose Graph View Transformation (GVT), a node- and feature-permutation-equivariant mapping in the view space. GVT serves as the building block for Recurrent GVT, a fully inductive model for node representation learning. Pretrained on OGBN-Arxiv and evaluated on 27 node-classification benchmarks, Recurrent GVT outperforms GraphAny, the prior fully inductive graph model, by +8.93% and surpasses 12 individually tuned GNNs by at least +3.30%. These results establish the view space as a principled and effective ground for fully inductive node representation learning. Code and datasets are available at `https://anonymous.4open.science/r/RGVT`.

## 1 Introduction

Foundation models in natural language processing (NLP) and computer vision (CV) have transformed how models are built and shared across datasets (Brown et al., 2020; Kolesnikov et al., 2020; Radford et al., 2021; Touvron et al., 2023). The common approach is to pretrain a backbone encoder on large-scale corpora and then adapt it to smaller datasets with lightweight, per-dataset predictors (He et al., 2016; Devlin et al., 2019). This strategy is especially effective when labeled data is scarce, as it leverages transferable knowledge accumulated during pretraining.

This paradigm is possible because models in these domains are trained and inferred on standardized input formats. In NLP, a tokenization process maps arbitrary text into embeddings from a shared vocabulary (Sennrich et al., 2015; Vaswani et al., 2017). In CV, images can be resized or patchified to a fixed resolution without semantic loss (Krizhevsky et al., 2012; Dosovitskiy et al., 2021). These conventions create a common representation space across datasets—tokens or pixels—enabling trained models to perform inference on unseen datasets directly, without any training.

Unlike text or images, graph-structured data such as social networks, molecular graphs, or knowledge graphs resist standardization into a fixed input format: both topology (numbers of nodes and edges) and feature specifications vary across graphs (Ribeiro et al., 2017; Pei et al., 2020; Platonov et al., 2023). Accordingly, designing a single graph model that can perform inference on previously unseen graphs is a non-trivial challenge. Graph neural networks (GNNs) address the variability in graph size by modeling graphs as node sets equipped with a connectivity structure and performing message passing over this structure (Kipf & Welling, 2017; Hamilton et al., 2017). Building on this, many works pursue stronger structural generalization, aiming to enhance a model's ability to operate on graphs with unseen connectivity patterns such as from homophily to heterophily (or vice versa) (Zeng et al., 2019; Qu et al., 2022; Zhao et al., 2023; Cantürk et al., 2024).

Despite recent advances on inductive applications for topology, inductive learning over arbitrary feature spaces remains under-explored. Most existing methods either assume a shared feature space across datasets or operate without node features. This is because almost all deep learning models, including GNNs, rely on feature transformations that assume a single, fixed feature space observed

during training, making them difficult to apply to graphs with incompatible feature spaces. Recent work (Zhao et al., 2025) defined this broader inductive generalization as *fully inductive* learning and introduced the first model capable of handling both arbitrary graph sizes and feature spaces. While novel, its architecture learns how to attend linear predictions rather than processing or transforming the graph itself, which limits its ability to serve as a representation function.

We propose *fully inductive node representation learning* (FI-NRL) as a new paradigm to address this limitation. The goal of FI-NRL is to train a function that produces node-representation matrices from graphs of arbitrary size and feature specification. A model that supports FI-NRL eliminates the need to retrain and tune graph models such as GNNs for every dataset, allowing a single model to operate across graphs with arbitrary numbers of nodes, edges, and feature spaces. The resulting representations can be cached and reused, while dataset-specific adaptation such as mapping to labels can be handled by lightweight predictors (e.g., small MLPs) that are cheap to train. This mirrors the common paradigm in NLP and CV, where a pretrained encoder provides reusable representations, and dataset-specific adaptation is handled by simple predictors.

In this work, we first formalize the FI-NRL problem and establish the conditions which a function is guaranteed to be fully inductive (Section 3). As the foundation of our framework, we introduce the *view space*, a novel representational axis for graphs induced jointly by their features and connectivity structure. Within this space, arbitrary graphs can be encoded as set of fixed-size *view vectors*, providing a standardized input format on which models can be trained and applied for inference. Building on this, we propose *Graph View Transformation* (GVT), which transforms a node-representation matrix through the view space while satisfying all conditions guaranteed to be fully inductive. We further provide a theoretical analysis comparing the expressivity of GVT with standard GNNs, and finally introduce Recurrent Graph View Transformation (RGVT), an architecture which serves as a fully inductive graph encoder for node classification.

We evaluate RGVT on node classification by pretraining it on OGBN-Arxiv (Hu et al., 2020a) and transferring it to 27 benchmark datasets spanning diverse sizes, domains, and feature types. Frozen RGVT backbone with a lightweight per-dataset predictor, the model outperforms GraphAny (Zhao et al., 2025) on 26 out of 27 benchmark datasets, achieving an average gains of +8.93%. Furthermore, RGVT surpasses 12 GNNs, despite each being individually tuned and trained for its target dataset. These results demonstrate that learning in the view space is not only theoretically grounded but also practically effective for fully inductive node representation learning (FI-NRL).

## 2 PRELIMINARIES

**Tensor Notations.** For a vector $\boldsymbol{a}$, let $a_i$ be its $i$-th element. For a matrix $\boldsymbol{A}$, $A_{i,j}$ denotes the $(i, j)$-th entry, while $\boldsymbol{A}_{i,:}$ and $\boldsymbol{A}_{:,j}$ denote its $i$-th row and $j$-th column, respectively. For a 3-D tensor $\mathbf{A}$, $\mathbf{A}_{i,j,k}$ denotes the element at position $(i, j, k)$, and $\mathbf{A}_{i,j,:}$ denotes the vector along the third mode.

**Graph Notations.** Let $\mathcal{G} = (V, E)$ be an undirected graph with node set $V$ and edge set $E$, where $N = |V|$ is the number of nodes. The adjacency matrix is denoted by $\boldsymbol{A} \in \{0, 1\}^{N \times N}$, where $A_{ij} = 1$ if and only if $(i, j) \in E$. The initial node-feature matrix is $\boldsymbol{X} \in \mathbb{R}^{N \times F}$. Each row $\boldsymbol{X}_{n,:}$ is a feature vector of node $n$, and each column $\boldsymbol{X}_{:,f}$ is a feature $f$ across nodes.

**Graph Neural Networks.** Given a graph $\mathcal{G}$, a graph neural network (GNN) iteratively updates the node-representation matrix $\boldsymbol{Z}^{(l)} \in \mathbb{R}^{N \times F_l}$ starting from the input node-feature matrix $\boldsymbol{Z}^{(0)} = \boldsymbol{X}$, where $F_l$ is the dimensionality of representation at layer $l$. Each layer applies two key operations, *aggregation* and *combination* (Gilmer et al., 2017; Xu et al., 2018b; Hu et al., 2020b):

$$\boldsymbol{Z}_{\mathcal{N}}^{(l)} = \text{Aggregate}^{(l)}(\boldsymbol{Z}^{(l-1)}, \boldsymbol{A}), \quad \boldsymbol{Z}^{(l)} = \text{Combine}^{(l)}(\boldsymbol{Z}_{\mathcal{N}}^{(l)}, \boldsymbol{Z}^{(l-1)}). \tag{1}$$

At each layer, the *aggregation* conveys structural information by propagation, multiplying the node-representation matrix $\boldsymbol{Z}^{(l)}$ with a matrix $\hat{\boldsymbol{A}} \in \mathbb{R}^{N \times N}$ derived from the adjacency matrix $\boldsymbol{A}$ either statically (Kipf & Welling, 2017; Hamilton et al., 2017) or dynamically (Veličković et al., 2018; Chien et al., 2021). The *combination* then integrates aggregated results and the original representation typically with a learnable matrix $\boldsymbol{W}^{(l)} \in \mathbb{R}^{F_{l-1} \times F_l}$ for feature transformation.

## 3    FULLY INDUCTIVE NODE REPRESENTATION LEARNING

We define *fully inductive node representation learning* (FI-NRL) as the task of learning a representation function $\Psi(\boldsymbol{X}, \boldsymbol{A}) = \boldsymbol{Z}$ that can map any graph of arbitrary number of nodes, edges, or features to a set of node representations. This formulation allows downstream, dataset-specific predictors to be trained directly on the resulting representations $\boldsymbol{Z}$, separating representation learning from task-specific modeling. Formally, a representation function $\Psi$ must satisfy

$$\Psi : \mathbb{R}^{N \times F} \times \{0, 1\}^{N \times N} \to \mathbb{R}^{N \times H} \quad \text{for all } N, F \geq 1,$$

requiring $\Psi$ to be operate on graphs with both arbitrary numbers of nodes $N$ and feature dimensionality $F$. We introduce two conditions that guarantee a function satisfies this requirement.

**R1. Node-permutation equivariance.** The function $\Psi$ must be equivariant to node permutations to ensure it respects graph isomorphisms. For any permutation matrix $\boldsymbol{P} \in \{0, 1\}^{N \times N}$:

$$\Psi(\boldsymbol{P}\boldsymbol{X}, \boldsymbol{P}\boldsymbol{A}\boldsymbol{P}^\top) = \boldsymbol{P}\,\Psi(\boldsymbol{X}, \boldsymbol{A}).$$

**R2. Feature-permutation equivariance.** To handle diverse and variable feature sets, $\Psi$ must also be equivariant to feature permutations. For any permutation matrix $\boldsymbol{Q} \in \{0, 1\}^{F \times F}$:

$$\Psi(\boldsymbol{X}\boldsymbol{Q}, \boldsymbol{A}) = \Psi(\boldsymbol{X}, \boldsymbol{A})\boldsymbol{Q}.$$

The two requirements are not only symmetry conditions. Together, they guarantee that the function $\Psi$ is well-defined regardless of the numbers of nodes $N$ and features $F$, as formalized below.

**Lemma 3.1** (Fully Inductive Graph Transformation)**.** *A function $\Psi$ that is equivariant to both node permutations (**R1**) and feature permutations (**R2**) is well-defined for graphs of arbitrary size $(N, F)$.*

The formal proof is provided in Appendix B. Intuitively, permutation equivariance enforces order-agnostic, set-based operations. Such operations naturally handle variable-sized inputs (e.g., a node's neighborhood), making the function independent of the number of elements—$N$ and $F$ in our case.

**R1** is a standard property satisfied by most graph models, including GNNs (Kipf & Welling, 2017; Hamilton et al., 2017; Veličković et al., 2018; Xu et al., 2018a; Maron et al., 2019), since they are explicitly designed to handle graphs of arbitrary size. In contrast, **R2** poses a key challenge that most existing GNN architectures fail to meet. This is because they encode knowledge through feature transformations that rely on learnable weight matrices, e.g., $\boldsymbol{W} \in \mathbb{R}^{F \times F'}$. Such matrices are tied to the feature size and its ordering, making the learned parameters invalid for graphs with different feature space. Together, R1 and R2 define the core challenge of FI-NRL: designing a function that produces node representations while agnostic to both graph size and feature dimensionality.

**Remark**   Although feature-permutation-invariant mappings can be fully inductive (Zhao et al., 2025; Finkelshtein et al., 2025; Eliasof et al., 2025), we assert that feature-permutation-equivariance is essential for effective representation learning. (1) Any invariant function is an invariant read-out of an underlying equivariant representation (see Proposition C.1), so equivariance offers strictly greater expressivity. (2) Invariant mappings discard feature-wise structure such as cross-channel interactions (see Lemma C.2), limiting the information available to downstream predictors. For these reasons, equivariance is necessary for learning rich fully inductive representations.

## 4    THE VIEW SPACE

We introduce the *view space*, a novel representation space in which graphs with arbitrary numbers of nodes, edges, and features are mapped into consistent, fixed-dimensional *view vectors*. We then propose *Graph View Transformation* (GVT), which is the parametric transformation satisfying dual permutation equivariance stated in Section 3 via the view space. Formal proofs of Lemma 4.2 and Theorem 4.7 in this section are provided in Appendix D.

### 4.1    UNCOVERING A HIDDEN SPACE OF GRAPHS

Graphs are conventionally represented by a node-feature matrix $\boldsymbol{X} \in \mathbb{R}^{N \times F}$. Each row $\boldsymbol{X}_{n,:}$ lies in the *feature space* $\mathbb{R}^F$, representing the features of node $n$, while each column $\boldsymbol{X}_{:,f}$ lies in the *node*

*space* $\mathbb{R}^N$, representing the feature $f$ across nodes. However, this two-dimensional representation cannot achieve simultaneous permutation equivariance across both axes: transformations along the feature dimension violate R2, while transformations along the node dimension violate R1.

Our key idea to overcome this limitation is to lift the representation into a higher-dimensional space by adding an extra axis, yielding a tensor $\mathbf{X} \in \mathbb{R}^{N \times F \times C}$. Then, the new axis $C$ permits transformations that remain equivariant under both node and feature permutations.

We introduce the *view space*, the third axis of the graph representation. This axis is formed through a *view stacking* operation, which collects multiple versions of the node-feature matrix and arranges them along this new dimension. Each version is produced by a *view finder*, with each view finder generating a distinct node-feature matrix that captures a different perspective of the graph.

**Definition 4.1** (View Finder). Given an adjacency matrix $\boldsymbol{A} \in \mathbb{R}^{N \times N}$, a **view finder** is a matrix-valued function $\nu : \mathbb{R}^{N \times N} \to \mathbb{R}^{N \times N}$ that satisfies node-permutation equivariance:

$$\nu(P\boldsymbol{A}P^\top) = P\,\nu(\boldsymbol{A})\,P^\top, \qquad \forall P \in \{0,1\}^{N \times N}.$$

Applying $\nu(\boldsymbol{A})$ to $\boldsymbol{X} \in \mathbb{R}^{N \times F}$ produces a propagated node-feature matrix $\nu(\boldsymbol{A})\boldsymbol{X} \in \mathbb{R}^{N \times F}$.

**Lemma 4.2.** *Common adjacency preprocessing methods used in GNNs—including self-augmented adjacency, degree-normalized variants (row-stochastic or symmetric) (Kipf & Welling, 2017; Hamilton et al., 2017), spectral and Laplacian filters (Defferrard et al., 2016), diffusion kernels (Gasteiger et al., 2019), and their polynomial extensions—are all valid view finders.*

As noted in Lemma 4.2, view finders are not new; they correspond to the adjacency preprocessing methods already used in the aggregation of GNNs. Prior studies show that different preprocessing choices emphasize different structural aspects of a graph, leading to distinct propagation outcomes and variations in performance across graphs (Mao et al., 2023; Subramonian et al., 2024).

From a set of distinct view finders, we obtain multiple node-feature matrices propagated differently, each capturing a distinct "view" of the graph. The *view stacking* operation then combines these matrices along an additional axis, producing a node-feature-view tensor $\mathbf{X} \in \mathbb{R}^{N \times F \times C}$.

**Definition 4.3** (View Stacking). Given adjacency matrix $\boldsymbol{A}$ and node-feature matrix $\boldsymbol{X}$, and a set of view finders $\{\nu_c\}_{c=1}^C$, the **view stacking** operation $\mathcal{V}$ constructs a node-feature-view tensor

$$\mathbf{X} = \mathcal{V}(\boldsymbol{X}, \boldsymbol{A} \mid \{\nu_c\}_{c=1}^C) = \left[\, \nu_1(\boldsymbol{A})\boldsymbol{X},\ \nu_2(\boldsymbol{A})\boldsymbol{X},\ \ldots,\ \nu_C(\boldsymbol{A})\boldsymbol{X} \,\right] \in \mathbb{R}^{N \times F \times C},$$

by stacking the C propagated node-feature matrices along a new axis of size C.

Introducing the new axis re-organizes the representation: each node-feature entry $(n, f)$ becomes a $C$-dimensional vector that captures how it appears across different views. The graph is therefore described by $N \times F$ such vectors, all sharing the same $C$-dimensional space. We call each vector a *view vector*, and the space they reside as *view space*, defined below:

**Definition 4.4** (View Vector). For each node-feature pair $(n, f)$ in the graph, the **view vector** $\boldsymbol{v}_{n,f} \in \mathbb{R}^C$ is the vector obtained by slicing the tensor $\mathbf{X}$ along its third dimension, i.e., $\boldsymbol{v}_{n,f} = \mathbf{X}_{n,f,:}$.

**Definition 4.5** (View Space). The **view space** is the vector space $\mathbb{R}^C$ in which the set of all view vectors $\{\boldsymbol{v}_{n,f}\}$ resides. Each coordinate axis in this space corresponds to a specific view finder.

Since the size $C$ and ordering of a view vector are determined solely by the predefined set of view finders, any graph—regardless of its number of nodes or features—can be represented in the view space by $N \times F$ view vectors of dimension $C$, as illustrated in Figure 1 (left). This provides graphs a standardized input format with shareable space, analogous to those in NLP and CV.

## 4.2 LEARNING REPRESENTATIONS VIA VIEW SPACE

To leverage this common space for representation learning, we introduce *Graph View Transformation* (GVT), a parametric function $\Psi$ that transforms the node-feature matrix $\boldsymbol{X}$ via the view space. GVT operates in two steps: (i) lifting the graph into the view space through view stacking, and (ii) applying a learnable mapping $\phi$ to collapse each view vector into a scalar, yielding the node-representation matrix $\boldsymbol{Z}$, as illustrated in Figure 1 (right). We formalize this as follows.

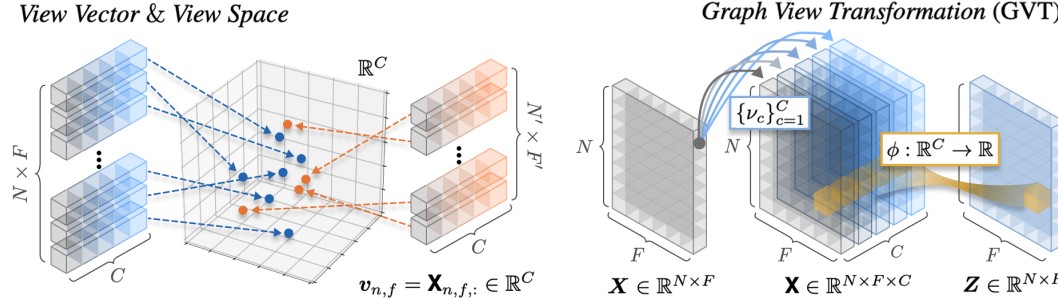

Figure 1: **(Left)** Graphs of varying sizes and feature dimensions can be mapped into the view space as $N \times F$ view vectors $\boldsymbol{v}_{n,f} = \mathbf{X}_{n,f,:} \in \mathbb{R}^C$. **(Right)** Graph View Transformation (GVT) transforms the node-feature matrix $\boldsymbol{X}$ into a node-feature-view tensor $\mathbf{X}$ through view stacking, and then applies $\phi$ to each view vector to produce a scalar, yielding the node-representation matrix $\boldsymbol{Z}$.

**Definition 4.6** (Graph View Transformation). Given an adjacency matrix $\boldsymbol{A}$, *Graph View Transformation* $\Psi$ maps a node-feature matrix $\boldsymbol{X}$ to a node-representation matrix $\boldsymbol{Z} \in \mathbb{R}^{N \times F}$ as

$$\Psi(\boldsymbol{X}, \boldsymbol{A}) = \left[ \phi(\mathbf{X}_{n,f,:} \mid \theta) \right]_{n \in [N], f \in [F]}, \quad \mathbf{X} = \mathcal{V}(\boldsymbol{X}, \boldsymbol{A} \mid \{\nu_c\}_{c=1}^C),$$

where $\mathcal{V}$ denotes view stacking using the view finder set $\{\nu_c\}_{c=1}^C$ from Definition 4.3, and $\phi : \mathbb{R}^C \to \mathbb{R}$ is a learnable dimension-collapsing function with parameters $\theta$.

The mapping $\phi$ takes each view vector $\mathbf{X}_{n,f,:}$ as input, learning patterns directly in the view space. Because $\phi$ is applied independently to each node-feature pair $(n, f)$, it is invariant to both node and feature order. Consequently, GVT satisfies dual permutation equivariance:

**Theorem 4.7** (Fully Inductive GVT). *GVT is equivariant to both node permutations (**R1**) and feature permutations (**R2**) required for fully inductive node representation learning (FI-NRL).*

The dimension-collapsing mapping $\phi$ can take any forms—such as linear projections, attention, or multi-layer perceptrons (MLPs)—while preserving the guarantees of Theorem 4.7. In this work, we use an MLP for its simplicity. Thus, GVT is a parametric transformation capable of handling graphs of any size and feature specification, fulfilling the requirements for FI-NRL.

## 5 UNDERSTANDING THE EXPRESSIVITY OF GVT

While GVT is provably fully inductive, its expressive power remains to be clarified. We analyze it by first examining the linear case and then extending to the nonlinear case, in direct comparison with GNNs. Formal proofs of Lemma 5.1 and Lemma 5.2 in this section are provided in Appendix D.

### 5.1 LINEAR GVT AS LEARNABLE AGGREGATION

We first consider the linear case of GVT which uses an affine function as the dimension-collapsing mapping: $\phi(\boldsymbol{v} \mid \boldsymbol{g}, b) = \boldsymbol{g}^\top \boldsymbol{v} + b$, with learnable parameters $\boldsymbol{g} \in \mathbb{R}^C$ and $b \in \mathbb{R}$. The linear mapping simplifies GVT to the following formulation:

$$\Psi(\boldsymbol{X}, \boldsymbol{A}) = [\boldsymbol{g}^\top \mathbf{X}_{n,f,:} + b]_{n \in [N], f \in [F]} = \left( \Sigma_{c=1}^C g_c \nu_c(\boldsymbol{A}) \right) \boldsymbol{X} + b \mathbf{1}_{N \times F}. \quad (2)$$

The resulting equation shows that the linear GVT combines adjacencies from different view finders using learned weights $g_c$ to form a new adjacency, which is then applied to the node-feature matrix. This formulation generalizes the static aggregation operations used in many GNNs.

**Lemma 5.1** (Generalization of Static Aggregations). *With a set of view finders including the identity and powers of the symmetric and row-normalized adjacency matrices, a linear GVT suffices to produce the aggregation operations of many GNNs (Kipf & Welling, 2017; Hamilton et al., 2017; Gasteiger et al., 2019; Wu et al., 2019; Chen et al., 2020; Chien et al., 2021).*

Thus, static aggregation can be viewed as a special case of linear GVT. This perspective explains why static aggregations (e.g., neighborhood averaging) apply consistently across arbitrary graphs—they are instances of GVT, which is fully inductive as shown in Section 4. Thus, learning a linear GVT can be seen as a data-driven search for an effective aggregation, and its inference on another graph amounts to adopting the learned knowledge of what constitutes an effective aggregation.

## 5.2 NONLINEAR GVT AS NODE-FEATURE DYNAMIC AGGREGATION

Introducing nonlinearity into GVT, such as using MLPs in the mapping $\phi$, enhances its expressivity by fundamentally altering the picture, as it can no longer be reduced to a weighted sum of adjacencies. To understand its behavior, we analyze it via local linear approximations.

**Lemma 5.2** (Local Linearization of Nonlinear GVT). *Suppose that $\phi : \mathbb{R}^C \to \mathbb{R}$ is twice differentiable almost everywhere. For any $\boldsymbol{v}_0 \in \mathbb{R}^C$, Taylor's theorem gives the local affine approximation*

$$\phi(\boldsymbol{v}) \ = \ g(\boldsymbol{v}_0)^\top \boldsymbol{v} + b(\boldsymbol{v}_0) + R_2(\boldsymbol{v}; \boldsymbol{v}_0),$$

*where $g(\boldsymbol{v}_0) = \nabla_{\boldsymbol{v}}\phi(\boldsymbol{v}_0)$ and $b(\boldsymbol{v}_0) = \phi(\boldsymbol{v}_0) - g(\boldsymbol{v}_0)^\top \boldsymbol{v}_0$. If $|\nabla^2\phi(\boldsymbol{v})| \leq M$ near $\boldsymbol{v}_0$, then $|R_2(\boldsymbol{v}; \boldsymbol{v}_0)| \leq \frac{M}{2}|\boldsymbol{v} - \boldsymbol{v}_0|^2$, so $\phi$ is locally affine with quadratic error decay.*

Thus, a nonlinear GVT behaves locally like a linear GVT around each view vector $\boldsymbol{v}_0$. In particular, the weights given by the gradient $g(\boldsymbol{v}_0) = \nabla_{\boldsymbol{v}}\phi(\boldsymbol{v}_0)$ depend on $\boldsymbol{v}_0$ unlike the static weights of the linear GVT globally applied to all node-feature pairs. This input dependence means that a nonlinear GVT implements an input-adaptive, or *dynamic*, aggregation in the sense of GNNs.

Each view vector corresponds to a specific node-feature pair $(n, f)$, allowing the nonlinear GVT to assign distinct aggregation behaviors at this fine-grained level. We refer to this property as *node-feature dynamic aggregation*. To the best of our knowledge, no existing GNNs achieve this granularity; prior work has been limited to node-wise (Zhang et al., 2021; 2022) or edge-wise (Veličković et al., 2018; Brody et al., 2021) dynamic aggregation. Thus, the nonlinear GVT is not only fully inductive but also unlocks new expressivity through fine-grained aggregation.

Consequently, the nonlinear GVT learns from each view vector—capturing how a node-feature pair varies across views—and determines its own aggregation. Inference of the trained GVT on a different graph is equivalent to adopting the learned mapping from view vectors to aggregation rules.

# 6 RGVT: A FULLY INDUCTIVE ENCODER FOR NODE CLASSIFICATION

Building on the fully inductive capability (Section 4) and expressivity (Section 5) of GVT, we take the next step toward realizing FI-NRL by introducing an architecture and training strategy that serves as a fully inductive graph encoder for node classification.

**Decoupling Depth and Paramterization.** Fully inductive capability does not guarantee coverage of high-order graph structures, as many graphs demand broader receptive fields to capture long-range dependencies (Chen et al., 2020; Alon & Yahav, 2021). However, creating a model by simply stacking GVT layers exhibits a key limitation: different graphs require different depths. Some tasks rely on local neighborhoods, others on multi-hop information, and the impact of over-smoothing varies across datasets. As a result, fixed-depth models often fail to generalize—even across graph datasets that share the same feature space (Xiao et al., 2023; Liu et al., 2024b).

To address this, we introduce *Recurrent Graph View Transformation* (RGVT), inspired by the principle of recurrent neural networks (RNNs) (Hochreiter & Schmidhuber, 1997) which decouple parameterization from sequence length. Instead of stacking multiple layers with distinct parameters, RGVT reuses a single nonlinear GVT $\Psi$ with shared parameters $\theta$ for $L$ times to yield

$$\boldsymbol{Z} = \Psi(\cdot, \boldsymbol{A} \mid \theta)^L(\boldsymbol{X}) \in \mathbb{R}^{N \times F}.$$

By decoupling parameterization from depth, the model can be applied at any $L$, allowing the depth to be selected per dataset without retraining. Recurrence expands the receptive field while adapting to dataset-specific requirements, ultimately improving generalization across diverse graphs.

**Training of RGVT.** Since RGVT is fully inductive by its architecture, it can be trained with any objective appropriate for a downstream task. We focus on node classification, which is commonly

used to evaluate the quality of node representations. We adopt an encoder-predictor paradigm for RGVT training; we attach a light-weight predictor $f_k$ for each dataset $k$ to adapt to its label space. For a graph dataset $\mathcal{G}_k$ with features $\boldsymbol{X}_k$, adjacency $\boldsymbol{A}_k$, labels $\boldsymbol{Y}_k \in \mathbb{R}^{N \times D_k}$ (where $D_k$ denotes the number of classes), recurrence depth $L_k$, and classification loss $\mathcal{L}$, the training objective is

$$\arg\min_{\hat{\theta}} \mathcal{L}\big(f_k\big(\Psi(\cdot, \boldsymbol{A}_k \mid \theta)^{L_k}(\boldsymbol{X}_k)\big), \boldsymbol{Y}_k\big), \quad f_k : \mathbb{R}^{N_k \times F_k} \to \mathbb{R}^{N_k \times D_k}.$$

The optimization target $\hat{\theta}$ includes $\{f_k, \theta\}$ during pretraining, and only $f_k$ during adaptation. This formulation enables pretrained RGVT to produce representations for arbitrary graphs without re-training, while predictors tailor it to each dataset. With an appropriate choice of recurrent depth $L_k$, lightweight predictors can adapt effectively to unseen graphs, even in label-scarce settings.

## 7 RELATED WORKS

**Graph Neural Networks.** GNNs have achieved notable success in graph learning by capturing structural information, but also face issues such as over-smoothing, over-squashing, and inconsistent performance across nodes, features, and graphs (Alon & Yahav, 2021; Rusch et al., 2023; Mao et al., 2023). To mitigate these, prior works explored structure editing (Jin et al., 2020; Ju et al., 2023), feature editing (Liu et al., 2021; Lee et al., 2025), and dynamic aggregation at the edge (Veličković et al., 2018; Brody et al., 2021) or node level (Zhang et al., 2021; 2022). Yet most approaches remain learning from the feature space, overlooking that aggregation is a view-space operation (Section 5). In contrast, GVT operates directly in the view space, leveraging its underexplored representational capacity. Experiments further show that RGVT outperforms 12 GNNs, demonstrating that structural understanding need not be mediated by features—the view space alone provides sufficient power.

**Graph Foundation Models.** Generalization across graphs with heterogeneous feature spaces has been explored in graph foundation models (GFMs). One direction is text-based alignment (Liu et al., 2024a; Chen et al., 2024a), where node features are treated as raw text or converted into natural language and then embedded using language models. This provides a shared textual space, but many graphs lack textual attributes, and converting numerical or categorical features into text often harms performance (Chen et al., 2024b). Another direction handles non-textual feature spaces by projecting features into a fixed-dimensional space using singular value decomposition paired with alignment modules (Yu et al., 2025; Zhao et al., 2024) or learnable patching (Sun et al., 2025). However, these methods require per-dataset fine-tuning for adaptation. Overall, while GFMs show promising generalization, neither directions supports fully inductive inference across arbitrary graphs.

**GraphAny.** GraphAny (Zhao et al., 2025) encodes knowledge in the relative-distance space of predictions, which is consistently defined across arbitrary graphs. By learning to attend over a set of linear, pseudo-inverse-based predictors, it becomes the first model to support fully inductive inference. Our work also targets fully inductive learning but differs in several key aspects: (1) We focus on representation learning, providing substantially higher expressivity; the resulting representations are reusable and allow flexible choice of downstream predictors while being decoupled from predictor quality. (2) Whereas GraphAny achieves full inductiveness via feature-permutation invariance, our framework is permutation-equivariant, which is advantageous for representation learning, as discussed in Section 3. (3) The linear predictors in GraphAny rely on static aggregation, whereas GVT performs dynamic aggregation, yielding greater expressivity in capturing structural patterns.

## 8 EXPERIMENTS

We evaluate how well RGVT generalizes and remains effective across graphs with diverse node and feature spaces. After pretraining on a large-scale dataset, we inference frozen RGVT to downstream graphs, training only a lightweight predictor with the recurrent depth $L$ chosen by validation accuracy. We consider two predictor variants: a linear classifier and a one-hidden-layer MLP.

**Datasets** We largely follow prior work (Zhao et al., 2025) in dataset selection and splitting. Pre-training is performed on the OGBN-Arxiv dataset (Hu et al., 2020a) using its public split. For downstream evaluation, we employ 27 node-classification benchmarks covering diverse graph structures and feature specifications. Public splits are used when available; otherwise, we sample 20 nodes per class for training, allocate half of the remainder for validation, and use the rest for testing. Given a

Table 1: Performance (%) comparison with predictor architectures and GraphAny across datasets grouped by feature type. **1st** and 2nd best results are highlighted. $\Delta$ reports relative improvements (%) over the baselines. RGVT consistently and significantly outperforms all baselines.

| Method | OGBN-Arxiv (1) | Signed Dense (5) | Unsigned Dense (4) | Sparse (4) | Binary Dense (3) | Binary Sparse (8) | One-hot (4) | Total Avg. (28) |
|---|---|---|---|---|---|---|---|---|
| Linear | $52.44_{\pm 0.04}$ | $53.29_{\pm 0.08}$ | $75.67_{\pm 0.64}$ | $66.41_{\pm 0.57}$ | $72.18_{\pm 0.69}$ | $57.11_{\pm 0.77}$ | $38.86_{\pm 2.52}$ | $59.41_{\pm 0.84}$ |
| MLP | $53.80_{\pm 0.14}$ | $55.08_{\pm 0.31}$ | $75.86_{\pm 0.68}$ | $69.02_{\pm 0.77}$ | $72.88_{\pm 0.64}$ | $57.65_{\pm 1.84}$ | $39.34_{\pm 2.43}$ | $60.43_{\pm 1.20}$ |
| GraphAny (Wisconsin) | $57.77_{\pm 0.45}$ | $59.12_{\pm 0.46}$ | $71.78_{\pm 1.22}$ | $81.61_{\pm 0.20}$ | $83.44_{\pm 0.24}$ | $55.25_{\pm 2.62}$ | $52.68_{\pm 0.40}$ | $64.72_{\pm 0.96}$ |
| GraphAny (Cora) | $58.58_{\pm 0.10}$ | $59.38_{\pm 0.42}$ | $71.76_{\pm 1.60}$ | $81.49_{\pm 0.10}$ | $83.35_{\pm 0.09}$ | $53.40_{\pm 2.44}$ | $53.30_{\pm 0.93}$ | $64.30_{\pm 0.94}$ |
| GraphAny (Arxiv) | $58.63_{\pm 0.14}$ | $59.70_{\pm 0.36}$ | $72.62_{\pm 1.32}$ | $81.68_{\pm 0.23}$ | $83.56_{\pm 0.08}$ | $54.18_{\pm 2.49}$ | $53.02_{\pm 0.35}$ | $64.71_{\pm 0.89}$ |
| GraphAny (Best) | $58.63_{\pm 0.14}$ | $59.74_{\pm 0.32}$ | $72.64_{\pm 1.02}$ | $81.89_{\pm 0.12}$ | $83.92_{\pm 0.13}$ | $55.58_{\pm 2.47}$ | $53.81_{\pm 0.77}$ | $65.30_{\pm 0.93}$ |
| **RGVT**+ Linear (Arxiv) | $70.14_{\pm 0.28}$ | $64.95_{\pm 0.44}$ | $76.44_{\pm 0.39}$ | **$84.33_{\pm 0.45}$** | **$85.11_{\pm 0.54}$** | $62.77_{\pm 1.93}$ | $58.85_{\pm 3.14}$ | $70.03_{\pm 1.26}$ |
| $\Delta$ Linear | ↑ 33.75% | ↑ 21.88% | ↑ 1.02% | ↑ 26.98% | ↑ 17.91% | ↑ 9.91% | ↑ 51.44% | ↑ 17.88% |
| $\Delta$ GraphAny (Best) | ↑ 19.63% | ↑ 8.72% | ↑ 5.23% | ↑ 2.98% | ↑ 1.42% | ↑ 12.94% | ↑ 9.37% | ↑ 7.24% |
| **RGVT**+ MLP (Arxiv) | **$71.11_{\pm 0.28}$** | **$66.37_{\pm 0.90}$** | **$77.12_{\pm 0.45}$** | $83.98_{\pm 0.81}$ | $84.86_{\pm 0.51}$ | **$63.87_{\pm 1.58}$** | **$62.48_{\pm 3.95}$** | **$71.13_{\pm 1.41}$** |
| $\Delta$ MLP | ↑ 32.17% | ↑ 20.50% | ↑ 1.66% | ↑ 21.67% | ↑ 16.44% | ↑ 10.79% | ↑ 58.82% | ↑ 17.71% |
| $\Delta$ GraphAny (Best) | ↑ 21.29% | ↑ 11.10% | ↑ 6.17% | ↑ 2.55% | ↑ 1.12% | ↑ 14.92% | ↑ 16.11% | ↑ 8.93% |

large number of benchmarks, the main paper reports results grouped by feature type. The full dataset list and grouping details are provided in Appendix L.

**Selection of View Finders.** For RGVT, we use $\{I\} \cup \{(D^{-1}A)^k, (D^{-\frac{1}{2}}AD^{-\frac{1}{2}})^k\}_{k=1}^{K}$ as the set of view finders, where $D$ is the degree matrix, and consider $K \in \{1, 2, 3\}$ as a hyperparameter. This choice is motivated by the fact that many aggregation operations used in popular GNNs can be expressed as linear combinations of elements in this set, and are therefore realizable by linear GVT, as shown in Lemma 5.1. Consequently, this set provides a rich aggregation space that the non-linear GVT can dynamically leverage. We present empirical analyses of the influence of the view-finder set on performance, along with guidelines for selecting candidate sets, in Appendix I.

**Implementation Details.** The search space further includes the number of MLP layers in the mapping $\phi$, the recurrent depth $L$ during pretraining, and the learning rate. Hyperparameters are chosen using validation accuracy on the pretraining dataset (OGBN-Arxiv) to ensure downstream graphs remain unseen. All experiments are repeated five times with independent seeds, and we report the mean and standard deviation. Detailed information is given in Appendix J.1.

**Baselines.** We evaluate three categories of baselines. (i) A linear classifier and a one-hidden-layer MLP, which serve as RGVT's predictors. (ii) GraphAny (Zhao et al., 2025), the prior fully inductive graph model, with three variants pretrained on the Wisconsin, Cora, and OGBN-Arxiv datasets, respectively. (iii) Twelve GNNs spanning diverse aggregation operations. Unlike RGVT and GraphAny, these models are trained and extensively tuned separately on each dataset, making them highly specialized. Further details of all baselines are provided in Appendix J.2.

## 8.1 Overall Performance

**Validation of FI-NRL.** In Table 1, we first validate whether fully inductive node representations produced by RGVT remain effective when adapted to graphs with entirely different structures and feature sets. Across all datasets, RGVT achieves substantial gains over its predictor models, outperforming the linear classifier by +17.71% and MLP by +17.88% on average. This demonstrates that the pretrained RGVT effectively integrates structural information into node representations, while being useful even when transferring from dense, signed word-embedding features (OGBN-Arxiv) to categorical, binary, or one-hot vectors. Overall, these results indicate that knowledge captured in the view space is not only theoretically but also empirically fully inductive.

We then compare RGVT with GraphAny, another fully inductive approach. As shown in Table 1, RGVT surpasses GraphAny even in its best-performing variant by +7.24% with the linear predictor and +8.93% with the MLP predictor on average. These results demonstrate that (1) the representation learning scheme of GVT is substantially more expressive than GraphAny's predictor-attention framework, and (2) GVT's dynamic aggregation behavior exploits structural information more effectively than GraphAny, whose linear predictors impose fixed, static aggregation. Moreover, RGVT maintains consistently strong performance across all feature subgroups, underscoring its robustness, whereas GraphAny exhibits larger fluctuations due to its linear predictors computed via pseudo-inverse, which are inherently sensitive to noise and feature type.

Table 2: Performance (%) comparison against 12 individually-trained GNN models across datasets grouped by feature type. 1st, 2nd, 3rd best results are highlighted. RGVT achieves the best performance on average, ranking 1st with the MLP predictor and 2nd with the linear predictor.

| Method | OGBN-Arxiv (1) | Signed Dense (5) | Unsigned Dense (4) | Sparse (4) | Binary Dense (3) | Binary Sparse (8) | One-hot (4) | Total Avg. (28) |
|---|---|---|---|---|---|---|---|---|
| GIN | $65.77_{\pm 1.08}$ | $55.57_{\pm 3.94}$ | $67.95_{\pm 3.90}$ | $71.93_{\pm 4.66}$ | $44.83_{\pm 4.72}$ | $43.30_{\pm 2.49}$ | $55.31_{\pm 3.88}$ | $54.98_{\pm 3.70}$ |
| GAT | $71.89_{\pm 0.24}$ | $60.89_{\pm 0.31}$ | $73.24_{\pm 0.62}$ | $82.50_{\pm 1.26}$ | $83.36_{\pm 0.90}$ | $49.20_{\pm 2.58}$ | $54.39_{\pm 4.98}$ | $63.88_{\pm 1.87}$ |
| GATv2 | $72.13_{\pm 0.11}$ | $64.45_{\pm 0.45}$ | $72.00_{\pm 1.22}$ | $81.68_{\pm 1.44}$ | $83.20_{\pm 0.80}$ | $49.61_{\pm 2.80}$ | $56.12_{\pm 3.41}$ | $64.57_{\pm 1.83}$ |
| S$^2$GC | $69.17_{\pm 0.02}$ | $59.94_{\pm 0.08}$ | $75.28_{\pm 0.35}$ | $82.04_{\pm 0.29}$ | $84.48_{\pm 0.14}$ | $52.28_{\pm 1.15}$ | $56.97_{\pm 1.35}$ | $65.31_{\pm 0.64}$ |
| SGC | $68.69_{\pm 0.03}$ | $58.95_{\pm 0.05}$ | $75.18_{\pm 0.26}$ | $82.36_{\pm 0.37}$ | $84.73_{\pm 0.10}$ | $51.15_{\pm 0.85}$ | $62.54_{\pm 3.24}$ | $65.66_{\pm 0.82}$ |
| JKNet | $71.73_{\pm 0.24}$ | $63.18_{\pm 1.19}$ | $76.12_{\pm 0.42}$ | $83.30_{\pm 0.71}$ | $84.02_{\pm 0.49}$ | $51.72_{\pm 1.60}$ | $58.46_{\pm 4.93}$ | $66.19_{\pm 1.59}$ |
| APPNP | $70.85_{\pm 0.15}$ | $64.86_{\pm 0.21}$ | $76.27_{\pm 0.36}$ | $83.66_{\pm 0.38}$ | $84.11_{\pm 0.21}$ | $55.21_{\pm 1.47}$ | $49.29_{\pm 1.26}$ | $66.26_{\pm 0.77}$ |
| GCN | $71.19_{\pm 0.30}$ | $61.49_{\pm 0.27}$ | $76.00_{\pm 0.38}$ | $83.48_{\pm 0.43}$ | $84.43_{\pm 0.26}$ | $53.03_{\pm 1.49}$ | $59.51_{\pm 1.43}$ | $66.46_{\pm 0.82}$ |
| GCNII | $72.05_{\pm 0.08}$ | $67.11_{\pm 0.25}$ | $77.54_{\pm 0.45}$ | $81.00_{\pm 0.97}$ | $83.65_{\pm 1.15}$ | $54.90_{\pm 1.73}$ | $56.11_{\pm 4.25}$ | $67.30_{\pm 1.47}$ |
| SAGE | $71.22_{\pm 0.24}$ | $67.98_{\pm 0.34}$ | $76.63_{\pm 0.65}$ | $82.14_{\pm 0.90}$ | $83.69_{\pm 0.51}$ | $60.07_{\pm 2.57}$ | $51.84_{\pm 3.39}$ | $68.36_{\pm 1.55}$ |
| GPRGNN | $69.45_{\pm 0.41}$ | $67.24_{\pm 0.28}$ | $76.78_{\pm 0.18}$ | $84.35_{\pm 0.40}$ | $85.26_{\pm 0.31}$ | $60.61_{\pm 1.65}$ | $50.40_{\pm 1.65}$ | $68.68_{\pm 0.94}$ |
| UniMP | $71.78_{\pm 0.12}$ | $69.09_{\pm 0.34}$ | $76.97_{\pm 0.60}$ | $82.28_{\pm 0.62}$ | $84.22_{\pm 0.54}$ | $59.15_{\pm 3.46}$ | $54.94_{\pm 3.61}$ | $68.86_{\pm 1.80}$ |
| **RGVT** + Linear | $70.14_{\pm 0.28}$ | $64.95_{\pm 0.44}$ | $76.44_{\pm 0.39}$ | $84.33_{\pm 0.45}$ | $85.11_{\pm 0.54}$ | $62.77_{\pm 1.93}$ | $58.85_{\pm 3.14}$ | $70.03_{\pm 1.26}$ |
| **RGVT** + MLP | $71.11_{\pm 0.28}$ | $66.37_{\pm 0.90}$ | $77.12_{\pm 0.45}$ | $83.98_{\pm 0.81}$ | $84.86_{\pm 0.51}$ | $63.87_{\pm 1.58}$ | $62.48_{\pm 3.95}$ | $71.13_{\pm 1.41}$ |

**Power of FI-NRL.** We next compare RGVT with 12 different GNNs. This comparison favors individual GNNs, as they are extensively tuned and trained on each downstream graph, whereas RGVT operates fully inductively, adapting only the recurrent depth $L$ and a lightweight predictor for each dataset. This setup enables us to assess whether fully inductive graph knowledge, transferred from graphs with completely different feature spaces, can stand against dataset-specific graph knowledge obtained through conventional GNN training.

Remarkably, RGVT with both predictors outperforms all 12 GNNs on average, as shown in Table 2. Moreover, RGVT with the MLP predictor surpasses the best-performing GNN, UniMP (Shi et al., 2021), by an average margin of +3.30%. This demonstrates that fully inductive graph knowledge in the view space can match or even exceed the dataset-specific knowledge obtained by GNNs. It also indicates that feature-space transformations are not essential for understanding graph—*graph view transformation* (GVT) provides a powerful alternative.

Finally, a Wilcoxon signed-rank test (Appendix A) further confirms the statistical significance of RGVT over each GNN across 28 datasets ($p < 0.05$), with GPRGNN (Chien et al., 2021) as the only exception ($p = 0.07$). Complete results are provided in Appendix M.

## 8.2 VISUALIZATION OF NODE-FEATURE DYNAMIC AGGREGATION

To demonstrate that the node-feature dynamic aggregation of nonlinear GVT described in Section 5 is not merely a theoretical property but an observable behavior, we compute the local linearizations defined in Lemma 5.2 and visualize them as a heatmap over sampled nodes and features in Figure 2:

$$\mathbf{H}_{n,f}^{(c)} = \left( \nabla_{\boldsymbol{v}} \phi(\mathbf{X}_{n,f,:}) \right)_c \in \mathbb{R}^C,$$

where $\mathbf{H}^{(c)}$ represents the 2-D heatmap for the $c$-th view. Each entry captures the contribution of the $c$-th view to a specific node-feature pair. In the figure, the $x$-axis is feature indices, the $y$-axis is node indices, and the color encodes the linearized weight. Distinct values across node-feature pairs confirm that the model carries out node-feature dynamic aggregation.

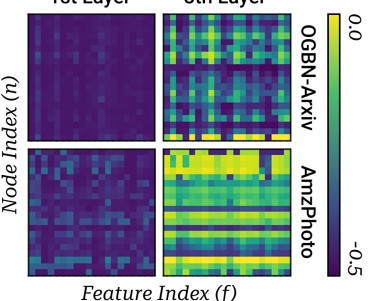

Figure 2: Node-feature heatmaps of linearly approximated aggregation weights for view from $\boldsymbol{A}_{\mathrm{SYM}}^2$.

This dynamicity persists from the pretraining graph (top: OGBN-Arxiv) to downstream graphs (bottom: AmzPhoto), showing that the mapping from view vectors to aggregation rules transfers. Across layers (left: 1st vs. right: 8th layer), the same node-feature pair receives different weights, indicating that recurrence does more than simply repeat— the aggregation adapts differently at each depth. The heatmap also reveals row- and column-wise patterns: entries from the same nodes or features behave similarly, even though $\phi$ processes each view vector $\mathbf{X}_{n,f,:}$ independently. The patterns suggests that view vectors carry shared semantics across nodes and features, while GVT implicitly captures.

Table 3: Ablation results (%) highlighting the contribution of nonlinearity and recurrence in RGVT.

| Method | OGBN-Arxiv (1) | Signed Dense (5) | Unsigned Dense (4) | Sparse (4) | Binary Dense (3) | Binary Sparse (8) | One-hot (4) | Total Avg. (28) |
|---|---|---|---|---|---|---|---|---|
| **RGVT** + MLP | $71.11_{\pm 0.28}$ | $66.37_{\pm 0.90}$ | $77.12_{\pm 0.45}$ | $83.98_{\pm 0.81}$ | $84.86_{\pm 0.51}$ | $63.87_{\pm 1.58}$ | $62.48_{\pm 3.95}$ | $71.13_{\pm 1.41}$ |
| w/o non-linearity | $70.22_{\pm 2.55}$ | $64.53_{\pm 2.67}$ | $75.89_{\pm 1.86}$ | $78.82_{\pm 5.99}$ | $84.16_{\pm 2.46}$ | $61.12_{\pm 5.66}$ | $56.13_{\pm 6.38}$ | $68.12_{\pm 4.39}$ |
| w/o recurrence | $70.91_{\pm 0.17}$ | $63.73_{\pm 0.43}$ | $73.79_{\pm 0.66}$ | $82.61_{\pm 0.79}$ | $83.90_{\pm 0.63}$ | $53.29_{\pm 1.92}$ | $54.53_{\pm 2.25}$ | $65.73_{\pm 1.22}$ |
| w/o both | $70.53_{\pm 0.25}$ | $61.69_{\pm 3.57}$ | $75.10_{\pm 0.43}$ | $77.52_{\pm 7.46}$ | $84.57_{\pm 0.53}$ | $53.41_{\pm 5.59}$ | $54.73_{\pm 1.86}$ | $64.96_{\pm 3.68}$ |

## 8.3 ABLATION STUDIES

In Table 3, we examine the role of two key components of RGVT: nonlinearity and recurrence. Removing nonlinearity reduces accuracy to 68.12%, comparable to GNN baselines. This is expected because without nonlinearity, GVT collapses into a learnable aggregation (Section 5), giving it expressive power fundamentally similar to that of standard GNNs. Nonlinearity is therefore essential, as it enables dynamic node-feature aggregation, having a representational capacity beyond existing GNNs and underpinning RGVT's performance. Eliminating recurrence leads to an even larger drop in accuracy, underscoring its role in receptive-field expansion and depth adaptation, both of which are critical for generalization across diverse graphs (Section 6). Further ablations on recurrent depth during pretraining and on the choice of pretraining dataset are reported in Appendix K.

## 9 CONCLUSION

In this work, we introduced the view space and Graph View Transformation (GVT) as a solution for fully inductive node representation learning (FI-NRL). By shifting the operational domain from the feature space to the view space, we established a unified ground where graphs of arbitrary size and feature specifications can be consistently represented and processed. This provides a standardized input space that allows a single model to be trained and applied across arbitrary graphs inductively. Theoretically, GVT subsumes standard GNN aggregation in its linear form, while nonlinearity provides greater expressivity through node-feature dynamic aggregation. Empirically, pretrained Recurrent GVT (RGVT) transferred the knowledge seamlessly across graphs with diverse feature types and outperformed 12 dataset-specialized GNNs. Together, these results show that GVT is not only theoretically grounded but also a practically effective framework for understanding graphs without feature-specific transformations, thereby achieving FI-NRL.

**Limitations and Future Works** While RGVT addresses FI-NRL effectively, we focus primarily on node classification. Extending the framework toward general-purpose representation learning—including different tasks such as link prediction and graph classification—is a promising direction toward graph foundation models. Furthermore, although the view space enables inductive inference across arbitrary graphs, effective generalization under structural and feature heterogeneity requires further improvements in the architecture and the training scheme. Finally, expanding the framework to incorporate edge features and to support broader graph families, such as heterogeneous graphs and hypergraphs, presents additional opportunities for advancement.

## ETHICS STATEMENT

In this paper, we aim to advance graph representation learning toward fully inductive applicability across graphs, with the primary goal of scientific research. Our approach is designed to broaden knowledge transfer across graphs, potentially benefiting a wide range of applications. We do not anticipate direct negative societal impacts, but we encourage responsible and ethical use of this work.

## REPRODUCIBILITY STATEMENT

To ensure reproducibility, we release both code and datasets as supplementary materials. Comprehensive details of hyperparameter settings, search spaces, and other experimental configurations are provided in Appendix J, together with the baseline search spaces. Dataset statistics are included in

Appendix L, and random split files are distributed with the released code. We also provide pretrained RGVT model checkpoints used to generate the exact results reported in this paper.

## USE OF LLMS

LLMs were used to polish the writing of the paper.

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

## A  WILCOXON SIGNED RANK TEST

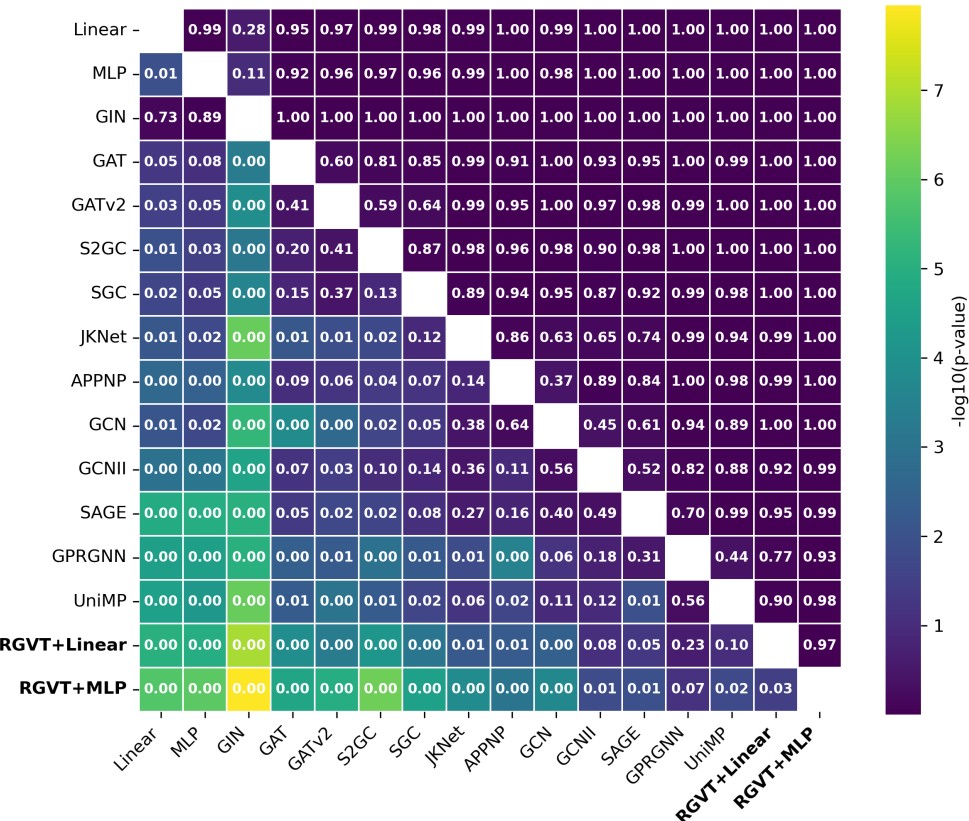

Figure 3:  Pairwise Wilcoxon signed-rank tests comparing RGVT using either a Linear or MLP predictor, against 12 fully tuned GNNs across 28 datasets. Brighter colors correspond to stronger statistical significance. Remarkably, RGVT with a MLP predictor—despite being pretrained on a single dataset—achieves statistically significant gains over all GNNs at the $p < 0.05$ level, with GPRGNN (Chien et al., 2021) as the sole exception ($p = 0.07$).

## B  PROOF OF LEMMA 3.1

**Lemma B.1** (Permutation equivariance $\Rightarrow$ node-size independence (root-preserving))**.** *For each $N \in \mathbb{N}$ let $\Psi_N : \mathbb{R}^{N \times F} \times \mathbb{R}^{N \times N} \to \mathbb{R}^{N \times G}$ satisfy*

$$\Psi_N(P\boldsymbol{X},\ P\boldsymbol{A}P^\top) = P\,\Psi_N(\boldsymbol{X}, \boldsymbol{A}) \quad \text{for all permutation matrices } P \in \{0, 1\}^{N \times N}.$$

*Fix a node $i$. Suppose that whenever two inputs $(\boldsymbol{X}, \boldsymbol{A})$ and $(\boldsymbol{X}', \boldsymbol{A}')$ have identical $i$-rooted neighborhoods as multisets,*

$$\left\{(\boldsymbol{X}_j, \boldsymbol{A}_{ij})\right\}_{j=1}^{N} \ = \ \left\{(\boldsymbol{X}'_j, \boldsymbol{A}'_{ij})\right\}_{j=1}^{N},$$

*there exists a permutation $P$ that bijects the two $i$-centered neighborhoods and fixes the root, i.e., $P(i) = i$. Then there exists a function $f$ on finite multisets, independent of $N$, such that*

$$\left(\Psi_N(\boldsymbol{X}, \boldsymbol{A})\right)_i = f\Big(\boldsymbol{X}_i,\ \left\{(\boldsymbol{X}_j, \boldsymbol{A}_{ij})\right\}_{j=1}^{N}\Big).$$

*In particular, the rule and its parameters do not depend on the node size $N$.*

*Proof.* Let $(\boldsymbol{X}, \boldsymbol{A})$ and $(\boldsymbol{X}', \boldsymbol{A}')$ be as in the statement. By the assumption, there is a permutation $P$ with $P(i) = i$ that maps the two $i$-rooted neighborhoods bijectively. By equivariance,

$$
\begin{aligned}
\big(\Psi_N(\boldsymbol{X}', \boldsymbol{A}')\big)_i &= \big(\Psi_N(P\boldsymbol{X}, \, P\boldsymbol{A}P^\top)\big)_i \\
&= \big(P\,\Psi_N(\boldsymbol{X}, \boldsymbol{A})\big)_i \\
&= \big(\Psi_N(\boldsymbol{X}, \boldsymbol{A})\big)_{P^{-1}(i)} \\
&= \big(\Psi_N(\boldsymbol{X}, \boldsymbol{A})\big)_i,
\end{aligned}
$$

where the last equality uses $P(i) = i$. Hence the $i$-th output depends only on the multiset $\{(\boldsymbol{X}_j, \boldsymbol{A}_{ij})\}_{j=1}^N$, not on labels or $N$. Define $f$ by evaluating $\Psi_N$ on any canonical ordering of that multiset and taking the $i$-th row; $f$ is well-defined and does not involve $N$. $\square$

**Lemma B.2** (Feature permutation equivariance $\Rightarrow$ feature-size independence (root-preserving)). *For each $F \in \mathbb{N}$ let $\Psi_F : \mathbb{R}^{N \times F} \times \mathbb{R}^{N \times N} \to \mathbb{R}^{N \times F}$ satisfy*

$$
\Psi_F(\boldsymbol{X}Q, \, \boldsymbol{A}) = \Psi_F(\boldsymbol{X}, \boldsymbol{A})\, Q \quad \text{for all feature permutations } Q \in \{0,1\}^{F \times F}.
$$

*Fix a feature index $f \in [F]$. Suppose that whenever two inputs $(\boldsymbol{X}, \boldsymbol{A})$ and $(\boldsymbol{X}', \boldsymbol{A})$ have identical column multisets $\{\boldsymbol{X}_{:,h}\}_{h=1}^F = \{\boldsymbol{X}'_{:,h}\}_{h=1}^F$, there exists a feature permutation $Q$ that bijects the two multisets and fixes $f$, i.e., $Q(f) = f$.*

*Then there exists a function $g$ on finite multisets, independent of $F$, such that*

$$
\big(\Psi_F(\boldsymbol{X}, \boldsymbol{A})\big)_{:,f} = g\Big(\boldsymbol{X}_{:,f}, \, \{\boldsymbol{X}_{:,h}\}_{h=1}^F, \, \boldsymbol{A}\Big).
$$

*In particular, the rule and its parameters do not depend on the feature dimension $F$.*

*Proof.* Let $(\boldsymbol{X}, \boldsymbol{A})$ and $(\boldsymbol{X}', \boldsymbol{A})$ be as in the statement. By assumption, there is $Q$ with $Q(f) = f$ that bijects the two column multisets.

By feature equivariance,

$$
\begin{aligned}
\big(\Psi_F(\boldsymbol{X}', \boldsymbol{A})\big)_{:,f} &= \big(\Psi_F(\boldsymbol{X}Q, \boldsymbol{A})\big)_{:,f} \\
&= \big(\Psi_F(\boldsymbol{X}, \boldsymbol{A})\, Q\big)_{:,f} \\
&= \big(\Psi_F(\boldsymbol{X}, \boldsymbol{A})\big)_{:,Q^{-1}(f)} \\
&= \big(\Psi_F(\boldsymbol{X}, \boldsymbol{A})\big)_{:,f}.
\end{aligned}
$$

where the last step uses $P(i) = i$ and $Q(f) = f$. Thus the $(i, f)$ entry depends only on the unordered $i$-rooted node multiset and on the unordered column multiset, not on labels, $N$, or $F$. Define $h$ by applying $\Psi_{N,F}$ after canonically ordering both multisets and reading the $(i, f)$ entry. $\square$

**Lemma B.3** ([Fully Inductive]{.underline} Graph Transformation). *A function $\Psi$ that is equivariant to both node permutations (**R1**) and feature permutations (**R2**) is well-defined for graphs of arbitrary size $(N, F)$.*

*Proof.* Let $\Psi_{N,F} : \mathbb{R}^{N \times F} \times \mathbb{R}^{N \times N} \to \mathbb{R}^{N \times G}$ satisfy R1 and R2 for every $(N, F)$.

**Step 1 (node-size independence).** Fix any node index $i \in [N]$. By Lemma B.1, there exists a function $f$ on finite multisets, independent of $N$, such that

$$
\big(\Psi_{N,F}(\boldsymbol{X}, \boldsymbol{A})\big)_i = f\Big(\boldsymbol{X}_i, \, \big\{(\boldsymbol{X}_j, \boldsymbol{A}_{ij})\big\}_{j=1}^N\Big).
$$

Hence the $i$-th row of $\Psi_{N,F}(\boldsymbol{X}, \boldsymbol{A})$ depends only on the $i$-rooted neighborhood multiset and not on $N$.

**Step 2 (feature-size independence).** Fix any feature index $f \in [F]$. By Lemma B.2, there exists a function $g$ on finite multisets, independent of $F$, such that

$$\big(\Psi_{N,F}(\boldsymbol{X}, \boldsymbol{A})\big)_{:,f} = g\Big(\boldsymbol{X}_{:,f}, \ \{\boldsymbol{X}_{:,h}\}_{h=1}^{F}, \ \boldsymbol{A}\Big).$$

Hence the $f$-th output channel depends only on the unordered multiset of feature columns (together with $\boldsymbol{A}$) and not on $F$.

**Step 3 (simultaneous independence of $N$ and $F$).** Let $(\boldsymbol{X}, \boldsymbol{A})$ and $(\boldsymbol{X}', \boldsymbol{A}')$ be two inputs (possibly with different sizes) and fix any pair $(i, f)$. Assume they have the same $i$-rooted neighborhood multiset and the same column multiset. By the root-preserving hypotheses in Lemmas B.1 and B.2, there exist permutations $P, Q$ with $P(i) = i$ and $Q(f) = f$ that biject these multisets. Using R1 and R2 (dual equivariance),

$$\begin{aligned}
\big(\Psi_{N',F'}(\boldsymbol{X}', \boldsymbol{A}')\big)_{i,f} &= \big(\Psi_{N,F}(P\boldsymbol{X}Q, \ P\boldsymbol{A}P^{\top})\big)_{i,f} \\
&= \big(P\,\Psi_{N,F}(\boldsymbol{X}, \boldsymbol{A})\,Q\big)_{i,f} \\
&= \big(\Psi_{N,F}(\boldsymbol{X}, \boldsymbol{A})\big)_{P^{-1}(i),\,Q^{-1}(f)} \\
&= \big(\Psi_{N,F}(\boldsymbol{X}, \boldsymbol{A})\big)_{i,f}.
\end{aligned}$$

Therefore, each entry $(i, f)$ depends only on (i) the *unordered* multiset of the $i$-rooted neighborhood and (ii) the *unordered* multiset of feature columns; it is independent of node/feature labels and of the sizes $(N, F)$.

From the three steps above, it follows that the entire output of $\Psi$ is well defined independently of $(N, F)$. In other words, $\Psi$ can be interpreted as a single rule for graphs of arbitrary size—one that assigns identical outputs to identical multisets—thereby the function is fully inductive.

$\square$

*Remark* B.4 (On the root-preserving hypothesis). The root-preserving assumption used in Lemmas B.1 and B.2 (i.e., choosing permutations $P, Q$ with $P(i) = i$ and $Q(f) = f$) is convenient but not necessary. One can remove it by *canonicalization*.

**Node side.** Fix $i$. Define a deterministic permutation $\kappa_i(\boldsymbol{X}, \boldsymbol{A}) \in \mathfrak{S}_N$ that (a) sends $i$ to index 1 and (b) sorts the $i$-rooted neighborhood multiset $\{(\boldsymbol{X}_j, \boldsymbol{A}_{ij})\}_{j=1}^{N}$ by a fixed total order (e.g., lexicographic in $(\boldsymbol{A}_{ij}, \boldsymbol{X}_j)$ with $j$ as a tie-breaker). Set

$$f\big(\boldsymbol{X}_i, \{(\boldsymbol{X}_j, \boldsymbol{A}_{ij})\}_j\big) := \Big(\Psi_N\big(\kappa_i(\boldsymbol{X}, \boldsymbol{A})\,\boldsymbol{X}, \ \kappa_i(\boldsymbol{X}, \boldsymbol{A})\,\boldsymbol{A}\,\kappa_i(\boldsymbol{X}, \boldsymbol{A})^{\top}\big)\Big)_1.$$

If two inputs share the same $i$-rooted multiset, then their canonicalized pairs coincide, and by node equivariance,

$$\big(\Psi_N(\boldsymbol{X}, \boldsymbol{A})\big)_i = \Big(\Psi_N\big(\kappa_i \boldsymbol{X}, \ \kappa_i \boldsymbol{A} \kappa_i^{\top}\big)\Big)_1.$$

Hence $f$ depends only on the multiset and is independent of $N$.

**Feature side.** Fix $f$. Define a deterministic permutation $\tau_f(\boldsymbol{X}) \in \mathfrak{S}_F$ that (a) sends $f$ to column 1 and (b) sorts the column multiset $\{\boldsymbol{X}_{:,h}\}_{h=1}^{F}$ by a fixed total order (lexicographic in the column vector with $h$ as a tie-breaker). Set

$$g\big(\boldsymbol{X}_{:,f}, \{\boldsymbol{X}_{:,h}\}_h, \boldsymbol{A}\big) := \Big(\Psi_F\big(\boldsymbol{X}\,\tau_f(\boldsymbol{X}), \ \boldsymbol{A}\big)\Big)_{:,1}.$$

If two inputs share the same column multiset, their canonicalized pairs coincide; by feature equivariance (with the chosen output action, e.g. $\rho_F(Q) = Q$ in the strong form),

$$\big(\Psi_F(\boldsymbol{X}, \boldsymbol{A})\big)_{:,f} = \Big(\Psi_F\big(\boldsymbol{X}\,\tau_f(\boldsymbol{X}), \ \boldsymbol{A}\big)\Big)_{:,1}.$$

Thus $g$ depends only on the multiset and is independent of $F$.

Combining the two sides gives the simultaneous independence of both $N$ and $F$ used in Theorem 4.7, without assuming root-preserving permutations.

## C ABOUT FEATURE PERMUTATION EQUIVARIANCE

### C.1 PERMUTATION INVARIANCE AND EQUIVARIANCE

In this section, we formalize a fact used in the main text: any feature-permutation–invariant predictor can be realized as an invariant readout of some feature-permutation–equivariant representation.

**Proposition C.1.** *Fix $N \geq 1$ and let $\mathfrak{S}_F$ be the permutation group of $F$ feature indices, acting on node-feature matrices $X \in \mathbb{R}^{N \times F}$ by right-multiplication $X \mapsto XQ$ with permutation matrices $Q \in \{0,1\}^{F \times F}$.*

*Let*
$$f : \mathbb{R}^{N \times F} \times \{0,1\}^{N \times N} \to \mathcal{Y}$$
*be* feature-permutation invariant, *i.e. it satisfies*
$$f(XQ, A) = f(X, A) \qquad \forall Q \in \mathfrak{S}_F, \ \forall (X, A).$$
*Then there exist*

- *a representation space $\mathcal{Z}$,*

- *a feature-permutation–equivariant map $\Phi : \mathbb{R}^{N \times F} \times \{0,1\}^{N \times N} \to \mathcal{Z}$,*

- *and a (not necessarily equivariant) post-map $\psi : \mathcal{Z} \to \mathcal{Y}$,*

*such that*
$$f = \psi \circ \Phi.$$
*In particular, every feature-permutation–invariant mapping can be written as an invariant readout of a feature-permutation–equivariant representation.*

*Proof.* We first introduce a simple equivalence relation on pairs $(X, A)$:
$$(X, A) \sim (X', A') \iff \exists Q \in \mathfrak{S}_F \text{ such that } X' = XQ \text{ and } A' = A.$$

Thus two graphs are equivalent if they only differ by a permutation of their feature indices. Let $(\mathbb{R}^{N \times F} \times \{0,1\}^{N \times N})/\sim$ denote the set of equivalence classes, and write $[(X, A)]$ for the class containing $(X, A)$.

Define the *quotient map*
$$\pi : \mathbb{R}^{N \times F} \times \{0,1\}^{N \times N} \to (\mathbb{R}^{N \times F} \times \{0,1\}^{N \times N})/\sim, \qquad \pi(X, A) = [(X, A)].$$
By construction we have
$$\pi(XQ, A) = [(XQ, A)] = [(X, A)] = \pi(X, A),$$
so $\pi$ is invariant under feature permutations.

We now equip the quotient space with the trivial action of feature permutations: for every $Q \in \mathfrak{S}_F$ and every class $[X, A]$, we set
$$Q \cdot [(X, A)] := [(X, A)].$$
With this action, $\pi$ becomes feature-permutation *equivariant* in the usual sense:
$$\pi(XQ, A) = [(XQ, A)] = [(X, A)] = Q \cdot [(X, A)] = Q \cdot \pi(X, A).$$

Next, since $f$ is invariant, it is constant on each equivalence class: if $(X', A') \sim (X, A)$, then $X' = XQ$ and $A' = A$ for some $Q$, hence
$$f(X', A') = f(XQ, A) = f(X, A).$$
Therefore there exists a unique function
$$\psi : (\mathbb{R}^{N \times F} \times \{0,1\}^{N \times N})/\sim \to \mathcal{Y}$$
such that
$$\psi([(X, A)]) = f(X, A),$$
and by definition we have $f = \psi \circ \pi$.

Finally, set $\mathcal{Z} := (\mathbb{R}^{N \times F} \times \{0,1\}^{N \times N})/\sim$ and $\Phi := \pi$. Then $\Phi$ is feature-permutation equivariant and $f = \psi \circ \Phi$, which proves the claim. $\square$

## C.2 Loss of Information

**Lemma C.2** (Permutation Invariance Removes Feature-Wise Structure). *Let*

$$g : \mathbb{R}^{N \times F} \times \{0, 1\}^{N \times N} \to \mathcal{Y}$$

*be feature-permutation invariant, i.e.,*

$$g(XQ, A) = g(X, A) \qquad \forall Q \in \mathfrak{S}_F, \ \forall (X, A), \tag{3}$$

*where $\mathfrak{S}_F$ is the permutation group on $F$ feature indices acting by right-multiplication $X \mapsto XQ$.*

*Then $g$ cannot represent any dependency or interaction tied to specific feature coordinates. In particular, for any indices $i \neq j$, let $X^{(i \leftrightarrow j)}$ denote the node-feature matrix obtained from $X$ by swapping its $i$-th and $j$-th feature columns. Then*

$$g(X, A) = g\big(X^{(i \leftrightarrow j)}, A\big) \qquad \forall (X, A). \tag{4}$$

*Proof.* Fix indices $i \neq j$ and let $Q_{(ij)} \in \mathfrak{S}_F$ be the permutation matrix that swaps feature indices $i$ and $j$ while leaving all other indices unchanged. For any $(X, A)$, define

$$X^{(i \leftrightarrow j)} := XQ_{(ij)}.$$

By construction, $X^{(i \leftrightarrow j)}$ is exactly $X$ with its $i$-th and $j$-th feature columns exchanged.

Since $g$ is feature-permutation invariant in the sense of equation 3, we have

$$g\big(X^{(i \leftrightarrow j)}, A\big) = g(XQ_{(ij)}, A) = g(X, A) \qquad \forall (X, A),$$

which proves equation 4. Thus $g$ assigns identical outputs to inputs that only differ by swapping the same pair of feature channels. $\square$

# D Proofs of Section 4

## D.1 Proof of Lemma 4.2

*Proof.* Let $A \in \mathbb{R}^{N \times N}$ be an adjacency matrix with degree matrix $D = \mathrm{diag}(A\mathbf{1})$, and let $P \in \{0, 1\}^{N \times N}$ be any permutation matrix. Each adjacency preprocessing operator listed in the lemma is a matrix-valued mapping $\nu : \mathbb{R}^{N \times N} \to \mathbb{R}^{N \times N}$ defined as follows:

1. **Self-augmented:** $\nu(A) = A + I$ (and trivially $\nu(A) = A$).

2. **Degree-normalized:** $\nu(A) := D^{-p}AD^{-q}$ for exponents $p, q \in \mathbb{R}$ (row/symmetric/column cases correspond to $p + q = 1$).

3. **Spectral/Laplacian filters:** $\nu(A) = f(L)$ with $L = D - A$ or $L_{\mathrm{sym}} = I - D^{-1/2}AD^{-1/2}$, for any matrix function $f$ given by a power series convergent on $\mathrm{spec}(L)$ (e.g., $e^{-tL}$).

4. **Diffusion kernels:** $\nu(A) = \alpha (I - (1 - \alpha)D^{-1}A)^{-1}$, with $\alpha \in (0, 1]$.

5. **Polynomial variants:** $\nu(A) = \sum_{k=0}^{K} c_k \nu_0(A)^k$ for some base operator $\nu_0$ from items 1–4.

We now show that each operator is permutation equivariant. Let $A \in \mathbb{R}^{N \times N}$ with degree matrix $D = \mathrm{diag}(A\mathbf{1})$, and let $P$ be a permutation matrix. For $A' = PAP^\top$ we have $D' = PDP^\top$. We use $P^\top = P^{-1}$, conjugation rules for sums/products, and $(PXP^\top)^{-1} = PX^{-1}P^\top$.

**1. Self-augmented.**

$$\nu(PAP^\top) = PAP^\top + I = P(A + I)P^\top = P\nu(A)P^\top.$$

**2. Degree-normalized.**

$$\nu(\boldsymbol{P}\boldsymbol{A}\boldsymbol{P}^\top) = (\boldsymbol{P}\boldsymbol{D}\boldsymbol{P}^\top)^{-p}\,(\boldsymbol{P}\boldsymbol{A}\boldsymbol{P}^\top)\,(\boldsymbol{P}\boldsymbol{D}\boldsymbol{P}^\top)^{-q} = \boldsymbol{P}\,\boldsymbol{D}^{-p}\,\boldsymbol{A}\,\boldsymbol{D}^{-q}\,\boldsymbol{P}^\top = \boldsymbol{P}\,\nu(\boldsymbol{A})\,\boldsymbol{P}^\top.$$

**3. Spectral/Laplacian filters.**

$$\boldsymbol{P}\boldsymbol{L}\boldsymbol{P}^\top = \boldsymbol{P}\boldsymbol{D}\boldsymbol{P}^\top - \boldsymbol{P}\boldsymbol{A}\boldsymbol{P}^\top = \boldsymbol{D}' - \boldsymbol{A}' = \boldsymbol{L}',$$

and for $f(z) = \sum_{k \geq 0} a_k z^k$,

$$f(\boldsymbol{P}\boldsymbol{L}\boldsymbol{P}^\top) = \sum_{k \geq 0} a_k (\boldsymbol{P}\boldsymbol{L}\boldsymbol{P}^\top)^k = \boldsymbol{P}\Big(\sum_{k \geq 0} a_k \boldsymbol{L}^k\Big)\boldsymbol{P}^\top = \boldsymbol{P}f(\boldsymbol{L})\boldsymbol{P}^\top.$$

Same for $\boldsymbol{L}_{\text{sym}}$.

**4. Diffusion kernels.** Let $\boldsymbol{B} = \boldsymbol{D}^{-1}\boldsymbol{A}$, so $\boldsymbol{B}' = (\boldsymbol{D}')^{-1}\boldsymbol{A}' = \boldsymbol{P}\boldsymbol{B}\boldsymbol{P}^\top$. Then

$$\nu(\boldsymbol{P}\boldsymbol{A}\boldsymbol{P}^\top) = \alpha(\boldsymbol{I} - (1-\alpha)\boldsymbol{B}')^{-1} = \alpha\,\boldsymbol{P}(\boldsymbol{I} - (1-\alpha)\boldsymbol{B})^{-1}\boldsymbol{P}^\top = \boldsymbol{P}\nu(\boldsymbol{A})\boldsymbol{P}^\top.$$

**5. Polynomial variants.** If $\nu_0(\boldsymbol{P}\boldsymbol{A}\boldsymbol{P}^\top) = \boldsymbol{P}\nu_0(\boldsymbol{A})\boldsymbol{P}^\top$, then

$$p(\nu_0(\boldsymbol{P}\boldsymbol{A}\boldsymbol{P}^\top)) = \sum_{k=0}^{K} c_k (\nu_0(\boldsymbol{P}\boldsymbol{A}\boldsymbol{P}^\top))^k = \sum_{k=0}^{K} c_k\,\boldsymbol{P}\nu_0(\boldsymbol{A})^k\boldsymbol{P}^\top = \boldsymbol{P}p(\nu_0(\boldsymbol{A}))\boldsymbol{P}^\top.$$

Thus, all listed preprocessing operators satisfy $\nu(\boldsymbol{P}\boldsymbol{A}\boldsymbol{P}^\top) = \boldsymbol{P}\nu(\boldsymbol{A})\boldsymbol{P}^\top$ and are permutation equivariant. □

D.2   PROOF OF THEOREM 4.7.

**Theorem D.1** (Fully Inductive GVT). *GVT is equivariant to both node permutations (**R1**) and feature permutations (**R2**) required for fully inductive node representation learning (FI-NRL).*

*Proof.* **Setup.** Let $\boldsymbol{X} \in \mathbb{R}^{N \times F}$, $\boldsymbol{A} \in \mathbb{R}^{N \times N}$, and view finders $\{\nu_c\}_{c=1}^{C}$. Define view stacking by

$$\mathbf{X} = \mathcal{V}(\boldsymbol{X}, \boldsymbol{A} \mid \{\nu_c\}) \quad \text{with} \quad \mathbf{X}_{:,:,c} = \nu_c(\boldsymbol{A})\,\boldsymbol{X} \in \mathbb{R}^{N \times F}.$$

Let the dimension-collapsing map be $\phi : \mathbb{R}^C \to \mathbb{R}$ with shared parameters $\theta$, applied independently at each $(n, f)$:

$$\boldsymbol{Z}_{n,f} = \phi(\mathbf{X}_{n,f,:} \mid \theta), \qquad \Psi(\boldsymbol{X}, \boldsymbol{A}) = \boldsymbol{Z}.$$

Each $\nu_c$ is node-permutation equivariant by the definition of view finder: for any permutation matrix $\boldsymbol{P} \in \{0,1\}^{N \times N}$, it holds that

$$\nu_c(\boldsymbol{P}\boldsymbol{A}\boldsymbol{P}^\top) = \boldsymbol{P}\,\nu_c(\boldsymbol{A})\,\boldsymbol{P}^\top. \tag{$\star$}$$

**R1: Node-permutation equivariance.** Let $\boldsymbol{P}$ be any node permutation. Consider inputs $(\boldsymbol{P}\boldsymbol{X}, \boldsymbol{P}\boldsymbol{A}\boldsymbol{P}^\top)$. For each $c$,

$$\mathbf{X}'_{:,:,c} = \nu_c(\boldsymbol{P}\boldsymbol{A}\boldsymbol{P}^\top)\,(\boldsymbol{P}\boldsymbol{X}) \overset{(\star)}{=} \boldsymbol{P}\,\nu_c(\boldsymbol{A})\,\boldsymbol{P}^\top\,\boldsymbol{P}\boldsymbol{X} = \boldsymbol{P}\big(\nu_c(\boldsymbol{A})\boldsymbol{X}\big) = \boldsymbol{P}\,\mathbf{X}_{:,:,c}.$$

Hence the stacked tensor transforms as $\mathbf{X}' = \boldsymbol{P}\,\mathbf{X}$ (mode-1 action). Therefore, for every $(n, f)$,

$$\mathbf{X}'_{n,f,:} = \mathbf{X}_{\pi(n),f,:} \quad \text{where } \pi \text{ is the permutation represented by } \boldsymbol{P}.$$

Since $\phi$ is applied identically and independently to each $(n, f)$ (shared $\theta$, no cross-index coupling),

$$\boldsymbol{Z}'_{n,f} = \phi(\mathbf{X}'_{n,f,:} \mid \theta) = \phi(\mathbf{X}_{\pi(n),f,:} \mid \theta) = \boldsymbol{Z}_{\pi(n),f},$$

i.e., $\boldsymbol{Z}' = \boldsymbol{P}\,\boldsymbol{Z}$. Thus

$$\Psi(\boldsymbol{P}\boldsymbol{X}, \boldsymbol{P}\boldsymbol{A}\boldsymbol{P}^\top) = \boldsymbol{P}\,\Psi(\boldsymbol{X}, \boldsymbol{A}),$$

which proves **R1**.

Table 4: Static GNN aggregations $p(\boldsymbol{A})$ reproduced by linear GVT through suitable coefficients $g_c$.

| Model | Aggregation $p(\boldsymbol{A})$ | Linear GVT coefficients |
|---|---|---|
| GCN (Kipf & Welling, 2017) | $\hat{\boldsymbol{A}}_{\text{SYM}}$ | $g = 1$ on $\hat{\boldsymbol{A}}_{\text{SYM}}$ |
| SAGE-mean (Hamilton et al., 2017) | $\boldsymbol{I}, \hat{\boldsymbol{A}}_{\text{RW}}$ | $g = 1$ on $\{\boldsymbol{I}, \hat{\boldsymbol{A}}_{\text{RW}}\}$ |
| SGC (Wu et al., 2019) | $\hat{\boldsymbol{A}}_{\text{SYM}}^{K}$ | $g = 1$ on $\hat{\boldsymbol{A}}_{\text{SYM}}^{K}$ |
| APPNP (Gasteiger et al., 2019) | $(1-\alpha) \sum_{k=0}^{K} \alpha^k \hat{\boldsymbol{A}}_{\text{SYM}}^{k}$ | $g_k = (1-\alpha)\alpha^k$ for $\hat{\boldsymbol{A}}_{\text{SYM}}^{k}$ |
| S$^2$GC (Zhu & Koniusz, 2021) | $\frac{1}{K} \sum_{k=1}^{K} \left((1-\alpha)\hat{\boldsymbol{A}}_{\text{SYM}}^{k} + \alpha\boldsymbol{I}\right)$ | $g_k = \frac{1-\alpha}{K}$ for $\hat{\boldsymbol{A}}_{\text{SYM}}^{k}$, $g = 1$ on $\boldsymbol{I}$ |
| GCNII (Chen et al., 2020) | $(1-\alpha)\hat{\boldsymbol{A}}_{\text{SYM}}, \alpha\boldsymbol{I}$ | $g = 1 - \alpha$ on $\hat{\boldsymbol{A}}_{\text{SYM}}$, $g = \alpha$ on $\boldsymbol{I}$ |
| GPRGNN (Chien et al., 2021) | $\sum_{k=0}^{K} \gamma_k \hat{\boldsymbol{A}}_{\text{SYM}}^{k}$ | $g_k = \gamma_k$ for $\hat{\boldsymbol{A}}_{\text{SYM}}^{k}$ |

**R2: Feature-permutation equivariance.** Let $\boldsymbol{Q} \in \{0,1\}^{F \times F}$ be a feature permutation. With inputs $(\boldsymbol{X}\boldsymbol{Q}, \boldsymbol{A})$, for each $c$,

$$\mathbf{X}''_{:,:,c} = \nu_c(\boldsymbol{A})\,(\boldsymbol{X}\boldsymbol{Q}) = \big(\nu_c(\boldsymbol{A})\boldsymbol{X}\big)\boldsymbol{Q} = \mathbf{X}_{:,:,c}\,\boldsymbol{Q}.$$

Hence $\mathbf{X}'' = \mathbf{X}\,\boldsymbol{Q}$ (mode-2 action), so for every $(n, f)$,

$$\mathbf{X}''_{n,f,:} = \mathbf{X}_{n,\sigma(f),:} \quad \text{where } \sigma \text{ is the permutation represented by } \boldsymbol{Q}.$$

Applying the same pointwise/shared $\phi$,

$$\boldsymbol{Z}''_{n,f} = \phi(\mathbf{X}''_{n,f,:} \mid \theta) = \phi(\mathbf{X}_{n,\sigma(f),:} \mid \theta) = \boldsymbol{Z}_{n,\sigma(f)},$$

i.e., $\boldsymbol{Z}'' = \boldsymbol{Z}\,\boldsymbol{Q}$. Therefore

$$\Psi(\boldsymbol{X}\boldsymbol{Q}, \boldsymbol{A}) = \Psi(\boldsymbol{X}, \boldsymbol{A})\,\boldsymbol{Q},$$

which proves **R2**.

**Conclusion.** Both **R1** and **R2** hold; hence the GVT layer is equivariant to node and feature permutations. $\qquad\square$

# E    Proofs of Section 5

## E.1    Proof of Lemma 5.1.

*Proof.* With view finders including $\boldsymbol{I}$ and powers of the normalized adjacencies $\{\hat{\boldsymbol{A}}_{\text{SYM}}^{k}, \hat{\boldsymbol{A}}_{\text{RW}}^{k}\}$ where $\hat{\boldsymbol{A}}_{\text{RW}} = \boldsymbol{D}^{-1}\boldsymbol{A}$ and $\hat{\boldsymbol{A}}_{\text{SYM}} = \boldsymbol{D}^{-\frac{1}{2}}\boldsymbol{A}\boldsymbol{D}^{-\frac{1}{2}}$ with $\boldsymbol{D}$ denoting the degree matrix, a linear GVT layer outputs

$$\boldsymbol{Z} = \sum_c g_c\,\nu_c(\boldsymbol{A})\,\boldsymbol{X} = p(\boldsymbol{A})\,\boldsymbol{X},$$

which matches many static aggregation filters $p(\boldsymbol{A})$ through suitable coefficients $\{g_c\}$. Representative choices of $\{g_c\}$ for well-known GNNs are summarized in Table 4.

Thus, each listed aggregation operator $p(\boldsymbol{A})$ is realized by a linear GVT. $\qquad\square$

## E.2    Proof of Theorem 5.2.

*Proof.* Assume $\nabla\phi$ is $M$-Lipschitz on a convex neighborhood of $v_0$ (equivalently, $\|\nabla^2\phi(w)\| \leq M$ for all $w$ on the segment $[v_0, v]$).

Fix $\boldsymbol{v}_0 \in \mathbb{R}^C$ and let $\boldsymbol{h} = \boldsymbol{v} - \boldsymbol{v}_0$. Consider the scalar function

$$\psi(t) = \phi(\boldsymbol{v}_0 + t\boldsymbol{h}), \qquad t \in [0, 1].$$

By the chain rule, $\psi'(t) = \nabla\phi(\boldsymbol{v}_0 + t\boldsymbol{h})^\top \boldsymbol{h}$ and $\psi''(t) = \boldsymbol{h}^\top \nabla^2\phi(\boldsymbol{v}_0 + t\boldsymbol{h})\,\boldsymbol{h}$ whenever the derivatives exist. By the fundamental theorem of calculus,

$$\phi(\boldsymbol{v}) - \phi(\boldsymbol{v}_0) = \psi(1) - \psi(0) = \int_0^1 \psi'(t)\,dt = \int_0^1 \nabla\phi(\boldsymbol{v}_0 + t\boldsymbol{h})^\top \boldsymbol{h}\,dt.$$

Table 5: Performance (%) comparison between GCN, GraphAny, and RGVT on six non-attributed graphs. 1st and 2nd best results for each dataset are highlighted. RGVT with DeepWalk embeddings achieves the best performance on 5 out of 6 datasets.

| | AirBrazil | | | AirEU | | | AirUSA | | |
|---|---|---|---|---|---|---|---|---|---|
| **Feature** | GCN | GraphAny | **RGVT** | GCN | GraphAny | **RGVT** | GCN | GraphAny | **RGVT** |
| One-hot | $63.08_{\pm 3.44}$ | $42.31_{\pm 0.00}$ | $\mathbf{66.15}_{\pm 10.67}$ | $41.50_{\pm 0.84}$ | $43.12_{\pm 0.76}$ | $47.00_{\pm 2.70}$ | $53.30_{\pm 1.18}$ | $46.31_{\pm 0.49}$ | $56.79_{\pm 2.26}$ |
| Random | $39.23_{\pm 5.70}$ | $37.69_{\pm 11.67}$ | $62.31_{\pm 7.05}$ | $41.38_{\pm 5.42}$ | $36.56_{\pm 2.37}$ | $\mathbf{47.25}_{\pm 3.98}$ | $49.12_{\pm 3.20}$ | $38.16_{\pm 1.71}$ | $54.09_{\pm 1.66}$ |
| DeepWalk | $36.15_{\pm 2.11}$ | $27.14_{\pm 0.48}$ | $\mathbf{72.31}_{\pm 4.49}$ | $43.50_{\pm 2.88}$ | $25.03_{\pm 0.07}$ | $\mathbf{48.00}_{\pm 2.22}$ | $57.12_{\pm 1.29}$ | $38.11_{\pm 1.98}$ | $\mathbf{58.27}_{\pm 1.10}$ |

| | Cora | | | Citeseer | | | Pubmed | | |
|---|---|---|---|---|---|---|---|---|---|
| **Feature** | GCN | GraphAny | **RGVT** | GCN | GraphAny | **RGVT** | GCN | GraphAny | **RGVT** |
| One-hot | $74.24_{\pm 2.29}$ | $65.40_{\pm 0.00}$ | $71.80_{\pm 1.07}$ | $52.04_{\pm 0.85}$ | $39.80_{\pm 0.00}$ | $46.04_{\pm 3.80}$ | $70.46_{\pm 0.65}$ | $61.30_{\pm 0.00}$ | $49.14_{\pm 6.15}$ |
| Random | $56.16_{\pm 1.51}$ | $22.18_{\pm 1.11}$ | $51.76_{\pm 2.98}$ | $36.48_{\pm 1.64}$ | $19.34_{\pm 1.47}$ | $33.90_{\pm 1.94}$ | $47.52_{\pm 2.10}$ | $35.88_{\pm 1.72}$ | $41.42_{\pm 2.84}$ |
| DeepWalk | $\mathbf{76.36}_{\pm 0.81}$ | $53.44_{\pm 1.94}$ | $\mathbf{74.96}_{\pm 0.94}$ | $52.36_{\pm 2.53}$ | $27.98_{\pm 1.79}$ | $\mathbf{53.20}_{\pm 0.67}$ | $\mathbf{74.20}_{\pm 1.73}$ | $64.12_{\pm 1.90}$ | $\mathbf{74.80}_{\pm 0.83}$ |

Add and subtract $\nabla\phi(\boldsymbol{v}_0)^\top \boldsymbol{h}$ inside the integral to obtain

$$\phi(\boldsymbol{v}) = \phi(\boldsymbol{v}_0) + \nabla\phi(\boldsymbol{v}_0)^\top \boldsymbol{h} + \int_0^1 \left( \nabla\phi(\boldsymbol{v}_0 + t\boldsymbol{h}) - \nabla\phi(\boldsymbol{v}_0) \right)^\top \boldsymbol{h} \, dt.$$

Define the remainder

$$R_2(\boldsymbol{v}; \boldsymbol{v}_0) = \int_0^1 \left( \nabla\phi(\boldsymbol{v}_0 + t\boldsymbol{h}) - \nabla\phi(\boldsymbol{v}_0) \right)^\top \boldsymbol{h} \, dt.$$

Using the mean value (integral) form with the Hessian and the bound on its operator norm,

$$\left\| \nabla\phi(\boldsymbol{v}_0 + t\boldsymbol{h}) - \nabla\phi(\boldsymbol{v}_0) \right\| \leq \int_0^t \left\| \nabla^2\phi(\boldsymbol{v}_0 + s\boldsymbol{h}) \right\| \|\boldsymbol{h}\| \, ds \leq Mt \|\boldsymbol{h}\|,$$

for all $t \in [0, 1]$ (here $\|\cdot\|$ is the Euclidean norm and $\|\nabla^2\phi\|$ its induced operator norm). Therefore

$$|R_2(\boldsymbol{v}; \boldsymbol{v}_0)| \leq \int_0^1 Mt \|\boldsymbol{h}\|^2 \, dt = \frac{M}{2} \|\boldsymbol{h}\|^2 = \frac{M}{2} \|\boldsymbol{v} - \boldsymbol{v}_0\|^2.$$

Finally, set $g(\boldsymbol{v}_0) = \nabla\phi(\boldsymbol{v}_0)$ and $b(\boldsymbol{v}_0) = \phi(\boldsymbol{v}_0) - g(\boldsymbol{v}_0)^\top \boldsymbol{v}_0$ to write

$$\phi(\boldsymbol{v}) = g(\boldsymbol{v}_0)^\top \boldsymbol{v} + b(\boldsymbol{v}_0) + R_2(\boldsymbol{v}; \boldsymbol{v}_0),$$

with the stated quadratic bound on $R_2$. This shows $\phi$ is locally affine with quadratic error decay near $\boldsymbol{v}_0$. □

## F  NON-ATTRIBUTED GRAPHS

In this section, we present additional experiments on non-attributed graphs. While three non-attributed datasets used in the main experiment (AirBrazil, AirEU, AirUSA) adopt one-hot vectors as node features, since the dimensionality of one-hot features scales with graph size, we further investigate alternative feature constructions to identify options that are scalable for RGVT.

**Datasets** We evaluate performance on six graphs: three airport traffic networks (AirBrazil, AirEU, AirUSA), which originally have no node features, and three citation networks (Cora, Citeseer, Pubmed), for which we explicitly remove the provided features. Inspired by prior work (Cui et al., 2022), we construct three types of artificial node features: (1) one-hot vectors based on node indices, (2) random Gaussian vectors, and (3) DeepWalk embeddings (Grover & Leskovec, 2016). For Deep-Walk, we adopt the hyperparameter settings from (Cui et al., 2022) consistently across datasets. For both the random and DeepWalk features, the feature dimension is fixed at 128.

Table 6: Performance (%) comparison with the MLP predictor and GraphAny across graph datasets with LLM-embedded features. **1st** and 2nd best results are highlighted. $\Delta$ reports relative improvements (%) over the baselines. RGVT shows consistent and significant gains across all datasets.

| | Cora | Citeseer | Pubmed | History | Children | Sportsfit | WikiCS |
|---|---|---|---|---|---|---|---|
| MLP | $78.54_{\pm1.24}$ | $72.73_{\pm0.95}$ | $84.75_{\pm1.01}$ | $71.64_{\pm3.76}$ | $31.73_{\pm1.24}$ | $75.14_{\pm2.10}$ | $74.69_{\pm0.85}$ |
| GraphAny | $79.27_{\pm0.63}$ | $74.15_{\pm0.35}$ | $78.69_{\pm0.04}$ | $45.87_{\pm0.13}$ | $25.77_{\pm0.16}$ | $79.27_{\pm0.02}$ | $71.01_{\pm0.24}$ |
| **RGVT + MLP** | $\mathbf{84.40_{\pm1.27}}$ | $\mathbf{75.22_{\pm1.66}}$ | $\mathbf{85.05_{\pm0.86}}$ | $\mathbf{75.86_{\pm1.44}}$ | $\mathbf{33.74_{\pm0.48}}$ | $\mathbf{82.82_{\pm2.23}}$ | $\mathbf{78.21_{\pm0.39}}$ |
| $\Delta$ MLP | ↑ **7.46**% | ↑ **3.42**% | ↑ **0.35**% | ↑ **5.89**% | ↑ **6.34**% | ↑ **10.23**% | ↑ **4.71**% |
| $\Delta$ GraphAny | ↑ **6.48**% | ↑ **1.44**% | ↑ **8.08**% | ↑ **65.38**% | ↑ **30.93**% | ↑ **4.49**% | ↑ **10.15**% |

Table 7: Performance (%) comparison with dataset-specialized GCN, GPRGNN, and UniMP on graph datasets with LLM-embedded features. Best results are highlighted in **bold**. RGVT achieves the highest accuracy on two datasets and remains competitive on the others.

| | Cora | Citeseer | Pubmed | History | Children | Sportsfit | WikiCS |
|---|---|---|---|---|---|---|---|
| GCN | $\mathbf{84.44_{\pm0.52}}$ | $75.65_{\pm0.99}$ | $83.46_{\pm0.33}$ | $72.58_{\pm4.88}$ | $33.85_{\pm1.84}$ | $82.23_{\pm2.47}$ | $78.70_{\pm0.91}$ |
| GPRGNN | $84.24_{\pm1.15}$ | $\mathbf{76.03_{\pm1.24}}$ | $84.59_{\pm1.58}$ | $73.38_{\pm4.70}$ | $35.70_{\pm0.82}$ | $81.92_{\pm3.00}$ | $\mathbf{79.83_{\pm1.00}}$ |
| UniMP | $83.97_{\pm0.81}$ | $75.71_{\pm1.15}$ | $84.71_{\pm1.27}$ | $75.05_{\pm2.11}$ | $\mathbf{36.23_{\pm3.17}}$ | $\mathbf{83.12_{\pm1.71}}$ | $78.51_{\pm0.66}$ |
| **RGVT + MLP** | $84.40_{\pm1.27}$ | $75.22_{\pm1.66}$ | $\mathbf{85.05_{\pm0.86}}$ | $\mathbf{75.86_{\pm1.44}}$ | $33.74_{\pm0.48}$ | $82.82_{\pm2.23}$ | $78.21_{\pm0.39}$ |

**Experiment Setting** The experimental setup follows the main experiment described in Appendix J. GCN is tuned and trained separately for each dataset and feature type, allowing it to specialize its optimization to each configuration. In contrast, GraphAny and RGVT use the pretrained weights from the main experiment (trained on OGBN-Arxiv) and are evaluated in a fully inductive manner. For RGVT, we use an MLP predictor in all experiments. All experiments are repeated five times with different random seeds, and we report the mean and standard deviation.

**Experiment Results** RGVT consistently outperforms GraphAny in 17 of the 18 dataset–feature configurations, demonstrating strong generalization to unseen, artificially constructed feature spaces. When equipped with DeepWalk embeddings, RGVT shows the highest accuracy on 5 of the 6 datasets and ranks second on the remaining one. These results highlight the importance of feature design for non-attributed graphs and show that DeepWalk embeddings, when combined with RGVT, yield strong and reliable performance.

# G  GRAPHS WITH LLM EMBEDDED FEATURES

In this section, we conduct additional experiments on graphs whose features are embedded using large language models (LLMs). The goal is to evaluate whether FI-NRL via RGVT, remains effective when node features are highly informative, such as those obtained from LLMs.

**Datasets** We adopt seven text-attributed graphs introduced in recent work (Wang et al., 2025), which investigates generalization on text-attributed graphs. Each node feature is generated using the `text-embedding-3-large` model from OpenAI, resulting in a 3704 dimensional embedding. The embedded text corresponds to the opening passages of academic papers (Cora, Citeseer, Pubmed), book descriptions or titles (History, Children), product titles in sports and fitness (Sports-Fit), and Wikipedia entries or article content (WikiCS). Dataset statistics are summarized in Table 8. For each dataset, we sample 20 nodes per class for training, 500 nodes for validation, and use the remaining nodes for testing. All experiments are repeated five times with different data splits.

**Experiment Setting** The experimental setup follows the main experiment described in Appendix J. We select GCN, GPRGNN, and UniMP—three models that perform strongly in our main experiments—along with MLP and GraphAny as baselines. While the GNNs baselines are tuned and trained separately for each dataset, GraphAny and RGVT use the pretrained weights from the main experiment (trained on OGBN-Arxiv) and are applied directly to these datasets in a fully inductive manner. For RGVT, we use the MLP as the predictor in all experiments.

**Experiment Results** Unlike in the main experiment, GraphAny fails to outperform the MLP. This indicates that the linear predictor in GraphAny cannot effectively exploit the rich semantic informa-

Table 8: Statistics of the seven graphs with features embedded by the LLM.

|  | Cora | Citeseer | Pubmed | History | Children | Sportsfit | WikiCS |
|---|---|---|---|---|---|---|---|
| #Nodes | 2708 | 3186 | 19717 | 41551 | 76875 | 173055 | 11701 |
| #Edges | 10556 | 8450 | 88648 | 503180 | 2325044 | 3020134 | 431726 |
| #Classes | 7 | 6 | 3 | 12 | 24 | 13 | 10 |
| Homophily Ratio | 0.8100 | 0.7841 | 0.8024 | 0.6398 | 0.4043 | 0.8980 | 0.6543 |
| Avg. Degree | 3.90 | 2.65 | 4.50 | 12.11 | 30.24 | 17.45 | 36.85 |

Table 9: Theoretical complexity of MLP, Feature Propagation, GCN, GraphAny, and RGVT, with scaling behavior for graph size and feature dimensionality. All methods scale linearly with graph size ($N, |E|$) and feature dimensionality ($F$). $L$ is the number of layers, $H$ is the hidden size, and $C$ is the number of linear GNNs (GraphAny) or view finders (RGVT).

| Method | End-to-end Complexity | Graph-Scale | Feature-Scale |
|---|---|---|---|
| MLP | $\mathcal{O}(NFH + (L-1)NH^2)$ | $\mathcal{O}(1)$ | $\mathcal{O}(F)$ |
| Feature Propagation | $\mathcal{O}(|E|F)$ | $\mathcal{O}(|E|)$ | $\mathcal{O}(F)$ |
| GCN | $\mathcal{O}(L|E|H) + \mathcal{O}(NFH + (L-1)NH^2)$ | $\mathcal{O}(|E|)$ | $\mathcal{O}(F)$ |
| GraphAny$^{(*)}$ | $\mathcal{O}(LC|E|F) + \mathcal{O}(NFH) + \mathcal{O}(NC^3)$ | $\mathcal{O}(|E|)$ | $\mathcal{O}(F)$ |
| RGVT | $\mathcal{O}(LC|E|F) + \mathcal{O}(LNFC^2)$ | $\mathcal{O}(|E|)$ | $\mathcal{O}(F)$ |

$(^*)$ GraphAny's pseudo-inverse step can scale quadratically with the feature dimension, up to $\mathcal{O}(F^2)$. However, we omitted here, as it can be effectively reduced through approximate solvers.

tion provided by LLM-embedded features. The degradation is more severe on the History, Children, and WikiCS datasets, which exhibit heterophilous structure. These results indicate further weak generalization to unseen graph structures, caused by the static aggregation of its linear predictors.

In contrast, Table 6 shows that RGVT consistently outperforms both baselines, yielding average gains of +5.48% over MLP and +18.14% over GraphAny. As shown in Table 7, RGVT also performs competitively against dataset-specialized GNNs, achieving the best accuracy on two datasets and remaining close to the top-performing model on the others. These results indicate that fully inductive graph knowledge remains effective even on graphs with rich semantic feature spaces. This effectiveness stems from RGVT's feature-permutation-equivariant design, which allows that feature-wise structure is preserved throughout transformation. As a result, the per-dataset MLP predictor can effectively leverage this information through the representations produced by RGVT.

Together, these results highlight GVT as an effective and robust solution for fully inductive node representation learning, without being constrained by the richness of the feature space.

# H  COMPLEXITY ANALYSIS

In this section, we analyze the computational characteristics of RGVT through theoretical complexity and empirical runtime measurements. We then discuss the practical efficiency benefits that arise from its fully inductive representation learning (FI-NRL) framework.

**Theoretical Analysis**  RGVT consists of a sequence of view-stacking operations followed by MLPs that map each view tensor to scalar. Since view stacking computes and concatenates $C$ propagated node-feature matrices, its cost is $C$ feature-propagation steps, i.e., $\mathcal{O}(C|E|F)$. The subsequent MLP transforms every node–feature pair, treating each view vector of dimension $C$ as a single input. Because the transformation operates on $NF$ node–feature pairs with input/hidden dimension $C$, this step incurs $\mathcal{O}(NFC^2)$ complexity. Thus, the total complexity of RGVT with recurrent depth $L$ is

$$\mathcal{O}\big(LC|E|F\ +\ LNFC^2\big),$$

which scales linearly with both the graph size and the feature dimension. We provides the summarizes result at Table 9 with complexity of GNN and GraphAny.

**Empirical Measurements**  We report the computational costs of RGVT with MLP predictor across three stages: (1) Pre-training: training on OGBN-Arxiv, (2) Adaptation: training an MLP predictor

Table 10: Time (s) for training on OGBN-Arxiv, adaptation across 27 datasets, and inference over all 28 datasets for RGVT, GraphAny, GCN, and GAT. Adaptation times for GNNs include full hyperparameter search under the search space used in the main experiment (see Appendix J), while values in parentheses denote training-only times for a 2-layer architecture.

| Method | Training (1) | Adaptation (27) | Inference (28) | Adaptation Tasks |
|---|---|---|---|---|
| GCN | 96.83 | 2day 10hr (55.28) | 0.002 | Hyperparameter search and training |
| GAT | 158.59 | 3day 23hr (91.30) | 0.003 | Hyperparameter search and training |
| GraphAny | 598.10 | 55.44 | 0.571 | Computes $C$ linear GNNs |
| **RGVT** + MLP | 607.31 | 151.24 | 0.052 | Trains $L$ MLP predictors |

on the $L$ representations produced at each depth across 27 datasets, and (3) Inference: inference RGVT to produce representations and performing MLP predictors' inference for all 28 datasets. For comparison, we also measure the corresponding computational costs for GCN, GAT, and GraphAny. All experiments are conducted on an NVIDIA RTX A6000 GPU.

As shown in Table 10, RGVT exhibits higher training and inference time than the GNN baselines due to three main reasons. First, RGVT uses a deeper architecture: with a recurrent depth of 8 in our configuration, its effective depth is roughly 4× that of 2-layer GNNs. Second, view stacking requires $C$ aggregations per layer, whereas standard GNNs perform only a single aggregation. Third, the dense transformation in GVT incurs $\mathcal{O}(NFC^2)$ complexity, compared to $\mathcal{O}(NH^2)$ in typical GNNs. Under our settings, when the feature dimension satisfies $F > 128^2/5^2 \approx 655$, the dense multiplication becomes slower by approximately a factor of $F/655$. Despite these factors, RGVT remains practical: its pre-training time on OGBN-Arxiv is approximately 10 minutes, and inference across all 28 graphs requires only 0.052 second.

**Practical Advantages** RGVT offers two significant complexity benefits. First, its fully inductive nature enables direct operation on unseen datasets without any retraining or fine-tuning. Only a lightweight predictor needs to be trained $L$ times to select an appropriate recurrent depth. However, GNNs require full training and extensive hyperparameter tuning per-dataset for sufficient performance. Second, once RGVT produces representations, they can be cached and reused. In contrast, GNNs must be retrained, and GraphAny must recompute its linear GNNs whenever the label space changes or when switching tasks. Together, these properties yield substantial efficiency benefits, mirroring the advantages of pretrained encoders in NLP and CV.

**Summary** Although RGVT has higher training and inference time than conventional GNNs, its overall complexity remains linear in both graph size and feature dimension. Crucially, RGVT operates without any retraining or tuning on new datasets, and its representations can be computed once and reused across different label spaces and tasks. These advantages make the FI-NRL paradigm and RGVT a compelling solution, as cross-dataset scalability and representation reusability are often the dominant factors for practical efficiency in real-world applications.

# I    GUIDELINES FOR SELECTING VIEW FINDER SETS

In this section, we analyze how different view finder choices affect RGVT's performance. We evaluate several graph convolution filters, varying hop sizes, and their combinations, and we provide practical guidance on how to construct effective candidate view finder sets.

**Experiment Configurations** We evaluate three families of view finders, each defined as a matrix-valued function of the adjacency matrix. The random-walk filter (RW) (Hamilton et al., 2017) and the symmetric normalized filter (SYM) (Kipf & Welling, 2017) are expressed as

$$\nu_{\text{rw}}^{(k)}(\boldsymbol{A}) = (\boldsymbol{D}^{-1}\boldsymbol{A})^k, \qquad \nu_{\text{sym}}^{(k)}(\boldsymbol{A}) = (\boldsymbol{D}^{-1/2}\boldsymbol{A}\boldsymbol{D}^{-1/2})^k,$$

where k denotes the number of propagation hops. We also include the Chebyshev polynomial basis (Cheb) (Defferrard et al., 2016), defined as

$$\nu_{\text{cheb}}^{(k)}(\boldsymbol{A}) = T_k(\tilde{\boldsymbol{L}}), \qquad \tilde{\boldsymbol{L}} = \frac{2\boldsymbol{L}}{\lambda_{\max}} - \boldsymbol{I}, \qquad \boldsymbol{L} = \boldsymbol{I} - \boldsymbol{D}^{-1/2}\boldsymbol{A}\boldsymbol{D}^{-1/2},$$

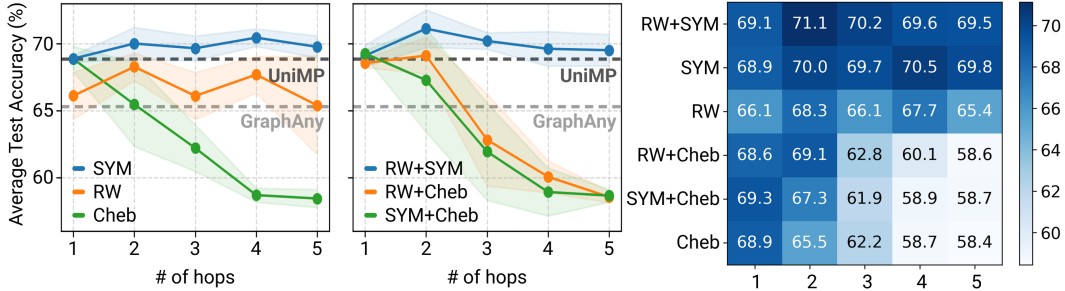

Figure 4: Average test accuracy (%) across 28 datasets using different view-finder sets for RGVT, shown in the plot (left) and heatmap (right). Symmetric normalization (SYM) and random-walk (RW) filters exhibit robust performance compared to the Chebyshev polynomial basis (Cheb), and their combination (RW+SYM) with moderate multi-hop yields the strongest results.

with the standard recurrence

$$T_0 = \boldsymbol{I}, \qquad T_1 = \tilde{\boldsymbol{L}}, \qquad T_k = 2\tilde{\boldsymbol{L}}T_{k-1} - T_{k-2} \quad (k \geq 2).$$

**Comparison of Filter Types** Our first set of experiments evaluates three view-finder sets, each consisting of one filter family applied over multiple hops:

$$\{\boldsymbol{I}\} \cup \{\nu_{\text{type}}^{(k)}(\boldsymbol{A})\}_{k=1}^{K},$$

where "type" denotes RW, SYM, or Cheb. As shown in Figure 4 (left), both RW and SYM maintain strong performance across hop sizes and consistently outperform GraphAny. While RW alone does not exceed the strongest GNN baseline (UniMP), SYM alone does, especially when multi-hop views are included. This suggests that symmetric normalization provides a more stable and informative view space, and that multi-hop views offer a richer set of signals for GVT to operate on.

In contrast, the Chebyshev filter performs well at $K = 1$ but degrades as more hops are added. We believe this behavior arises because higher-order Chebyshev polynomials increasingly amplify high-frequency components, introducing more noise into the view space. Without any mechanism to suppress this noise, GVT may overfit these noisy signals during training, leading to reduced performance. These observations suggest that, to leverage multi-hop information in the view space, smoother diffusion-style filters (RW, SYM) are more stable than Chebyshev filters.

**Comparison of Filter-Type Combinations** We then study whether mixing different filter families provides additional benefits. Each view-finder set includes two filter types (RW-SYM, RW-Cheb, SYM-Cheb), again evaluated across multiple hops:

$$\{\boldsymbol{I}\} \cup \{\nu_{\text{type1}}^{(k)}(\boldsymbol{A})\}_{k=1}^{K} \cup \{\nu_{\text{type2}}^{(k)}(\boldsymbol{A})\}_{k=1}^{K}.$$

Mixing RW and SYM filters improves performance for hop sizes 1 through 3 but not for 4 and 5. This suggests that multiple filter types can introduce complementary information into the view space, yielding better performance at moderate hop sizes. However, at larger hops the benefit diminishes, likely due to redundant or overly diffuse information.

Consistent with the earlier results, mixing Cheb with either RW or SYM does not improve performance beyond $K = 1$. This suggests that the degradation caused by high-frequency noise in the view space cannot be mitigated simply by adding more stable views.

**Guidelines** Based on these observations, we recommend constructing view-finder sets by combining diffusion-style filters (SYM, RW) with moderate hop sizes (typically 2–3). Spectral filters such as Chebyshev should be used cautiously, as higher-order variants can amplify high-frequency noise and lead to overfitting in GVT. Finally, because the computational cost of RGVT grows quadratically with the number of view finders (see Appendix H), it is important to keep the set compact.

## J EXPERIMENT CONFIGURATIONS

### J.1 OUR METHOD

For RGVT, we use $\{\boldsymbol{I}\} \cup \{(\boldsymbol{D}^{-1}\boldsymbol{A})^k, (\boldsymbol{D}^{-\frac{1}{2}}\boldsymbol{A}\boldsymbol{D}^{-\frac{1}{2}})^k\}_{k=1}^{K}$ as the view-finder set, where $\boldsymbol{D}$ is the degree matrix and consider $K \in \{1, 2, 3\}$ as a hyperparameter. The search space further includes the number of MLP layers in the mapping $\phi$ of GVT $D \in \{1, 2, 3\}$, the recurrent depth $L \in \{2, 4, 6, 8\}$ during pretraining, and the learning rate $\eta \in \{0.01, 0.05\}$. We use Gaussian Error Linear Units (GELUs) as the activation function in the nonlinear GVT.

Hyperparameters are selected using only the pretraining dataset. After pretraining is completed, we reinitialize the predictor, retrain the predictor on the same dataset while keeping RGVT frozen, and evaluate validation accuracy. This procedure offers a more reliable basis for selecting a model that generalizes well than relying on the validation accuracy recorded during pretraining.

The selected hyperparameter settings from the search are as follows:

- RGVT + Linear : $K = 2, D = 2, L = 8, \eta = 0.01$,
- RGVT + MLP : $K = 2, D = 2, L = 8, \eta = 0.005$.

### J.2 BASELINES

For the GNN baselines(Kipf & Welling, 2017; Hamilton et al., 2017; Xu et al., 2018a;b; Veličković et al., 2018; Wu et al., 2019; Gasteiger et al., 2019; Chen et al., 2020; Brody et al., 2021; Zhu & Koniusz, 2021; Chien et al., 2021; Shi et al., 2021) , we conducted a hyperparameter search over hidden dimensions {64, 128, 256}, depths {1,2,3,4,5}, and learning rates {0.01, 0.005, 0.001}, performed separately for each dataset. For GPRGNN and JKNet, we use the official implementation from the GPRGNN public repository (`https://github.com/jianhao2016/GPRGNN`), while all other models are adopted from PyG (Fey & Lenssen, 2019) implementation.

Both GNN baselines and RGVT are optimized with the Adam optimizer for up to 2500 epochs, with early stopping if no improvement in validation accuracy is observed for 200 consecutive epochs. This setup follows prior work (Luo et al., 2024), which demonstrated that extensive hyperparameter search leads to strong GNN performance. All hyperparameter searches, both for RGVT and for the GNN baselines are performed over 5 independent runs, and final test results are re-evaluated under the configuration selected based on average validation accuracy.

## K ADDITIONAL ABLATION STUDIES

We conducted two additional ablation studies on (i) the recurrent depth of RGVT during pretraining and (ii) the choice of pretraining dataset. First, as shown in Table 11, RGVT generally exhibits improved performance as the recurrent depth increases. Although the main experiment's search space did not include the best-performing depth, this suggests that a more extensive hyperparameter search could further enhance RGVT's performance. We attribute these gains to two factors: deeper pretraining allows the model to experience more diverse feature distributions, and during adaptation, a finer-grained choice of depth enables more compatible transfer.

Second, we evaluated three alternative datasets for pretraining. RGVT consistently performed worse on these datasets compared to OGBN-Arxiv, suggesting that it benefits from the larger scale of OGBN-Arxiv. Unlike GraphAny, whose performance remained stable across different datasets, RGVT showed sensitivity to pretraining data choice. This highlights the importance of pretraining dataset selection, and we leave systematic exploration of dataset choice and the construction of dedicated pretraining corpora for future work.

## L DATASET DETAILS

Our dataset selection and splitting strategy largely follows prior work GraphAny (Zhao et al., 2025). We access datasets through PyG (Fey & Lenssen, 2019), DGL (Wang et al., 2019), and the heterophilous graph collection from the Yandex-Research repository `https://github.com/`

Table 11: Performance (%) of RGVT + MLP under different recurrent depths $L$ during pretraining. The **best** and second-best results are highlighted.

| Depths $(L)$ | Ogbn-arxiv (1) | Signed Dense (5) | Unsigned Dense (4) | Sparse (4) | Binary Dense (3) | Binary Sparse (8) | One-hot (4) | Total Avg. (28) |
|---|---|---|---|---|---|---|---|---|
| 24 | $71.11_{\pm0.03}$ | $66.52_{\pm0.87}$ | $77.26_{\pm0.43}$ | $83.80_{\pm1.62}$ | $84.88_{\pm0.41}$ | $63.74_{\pm2.78}$ | $66.15_{\pm5.31}$ | $71.64_{\pm2.05}$ |
| 20 | $71.32_{\pm0.10}$ | **$66.60_{\pm0.64}$** | **$77.64_{\pm0.24}$** | $83.80_{\pm0.54}$ | **$85.24_{\pm0.47}$** | $64.07_{\pm2.62}$ | **$68.25_{\pm4.74}$** | **$72.14_{\pm1.70}$** |
| 16 | **$71.38_{\pm0.10}$** | $66.20_{\pm0.69}$ | $77.35_{\pm0.37}$ | $83.13_{\pm1.62}$ | $84.89_{\pm0.64}$ | **$64.88_{\pm2.23}$** | $67.64_{\pm4.49}$ | $72.04_{\pm1.75}$ |
| 12 | $71.28_{\pm0.28}$ | $66.13_{\pm1.03}$ | $77.50_{\pm0.57}$ | $83.39_{\pm1.18}$ | $84.94_{\pm0.57}$ | $64.07_{\pm1.56}$ | $65.55_{\pm3.82}$ | $71.57_{\pm1.49}$ |
| 8 | $71.11_{\pm0.28}$ | $66.37_{\pm0.90}$ | $77.12_{\pm0.45}$ | **$83.98_{\pm0.81}$** | $84.86_{\pm0.51}$ | $63.87_{\pm1.58}$ | $62.48_{\pm3.95}$ | $71.13_{\pm1.41}$ |
| 4 | $71.26_{\pm0.19}$ | $66.32_{\pm0.63}$ | $76.68_{\pm0.37}$ | $82.85_{\pm2.24}$ | $84.70_{\pm0.55}$ | $62.02_{\pm2.48}$ | $55.93_{\pm3.59}$ | $69.42_{\pm1.77}$ |
| 2 | $71.22_{\pm0.15}$ | $65.09_{\pm2.00}$ | $75.14_{\pm1.65}$ | $83.11_{\pm0.73}$ | $84.51_{\pm0.99}$ | $56.02_{\pm4.70}$ | $56.25_{\pm3.65}$ | $67.32_{\pm2.67}$ |

Table 12: Performance (%) of RGVT + MLP under different pretraining datasets. Pretraining on OGBN-Arxiv provides consistent improvements across all feature types and achieves the highest overall average performance compared to smaller datasets (AmzRatings, Cora, Wisconsin).

| Method | Ogbn-arxiv (1) | Signed Dense (5) | Unsigned Dense (4) | Sparse (4) | Binary Dense (3) | Binary Sparse (8) | One-hot (4) | Total Avg. (28) |
|---|---|---|---|---|---|---|---|---|
| Ogbn-Arxiv | **$71.11_{\pm0.28}$** | **$66.37_{\pm0.90}$** | **$77.12_{\pm0.45}$** | **$83.98_{\pm0.81}$** | **$84.86_{\pm0.51}$** | **$63.87_{\pm1.58}$** | $62.48_{\pm3.95}$ | **$71.13_{\pm1.41}$** |
| AmzRatings | $69.09_{\pm0.52}$ | $66.24_{\pm0.69}$ | $76.37_{\pm0.95}$ | $82.90_{\pm1.18}$ | $83.55_{\pm1.54}$ | $57.72_{\pm4.12}$ | $58.23_{\pm7.05}$ | $68.34_{\pm2.78}$ |
| Cora | $68.31_{\pm4.96}$ | $64.94_{\pm3.01}$ | $75.41_{\pm1.31}$ | $79.42_{\pm7.97}$ | $82.93_{\pm3.22}$ | $54.21_{\pm5.47}$ | $61.07_{\pm8.53}$ | $66.81_{\pm4.99}$ |
| Wisconsin | $68.06_{\pm3.09}$ | $64.64_{\pm2.08}$ | $75.44_{\pm1.60}$ | $79.78_{\pm4.39}$ | $82.13_{\pm2.88}$ | $57.33_{\pm5.88}$ | $65.46_{\pm7.08}$ | $68.25_{\pm4.23}$ |

`yandex-research/heterophilous-graphs`. For the Chameleon and Squirrel datasets, we adopt the filtered versions provided in this repository, as the original datasets have been reported to contain issues (Platonov et al., 2023). Detailed statistics for each dataset, together with their assigned feature-type groups used in the main text, are summarized in Table 13.

Table 13: Statistics of the 28 node-classification benchmarks used in this work, spanning diverse domains, feature specifications and structural properties.

| Dataset | #Nodes | #Edges | #Features | #Classes | Feature Type | Train/Val/Test Ratios (%) | #Train Nodes | #Val Nodes | #Test Nodes | Homophily Ratio | Avg. Degree |
|---|---|---|---|---|---|---|---|---|---|---|---|
| AirBrazil | 131 | 2137 | 131 | 4 | One-hot | 61.07/19.08/19.85 | 80 | 25 | 26 | 0.4307 | 16.31 |
| Texas | 183 | 741 | 1703 | 5 | Binary Sparse | 47.54/31.69/20.22 | 87 | 58 | 37 | 0.0609 | 4.05 |
| Cornell | 183 | 737 | 1703 | 5 | Binary Sparse | 47.54/32.24/20.22 | 87 | 59 | 37 | 0.1227 | 4.03 |
| Wisconsin | 251 | 1151 | 1703 | 5 | Binary Sparse | 47.81/31.87/20.32 | 120 | 80 | 51 | 0.1778 | 4.59 |
| AirEU | 399 | 12385 | 399 | 4 | One-hot | 20.05/39.85/40.10 | 80 | 159 | 160 | 0.4046 | 31.04 |
| Chameleon | 890 | 18598 | 2325 | 5 | Binary Sparse | 45.96/32.25/21.80 | 409 | 287 | 194 | 0.2361 | 20.9 |
| AirUSA | 1190 | 28388 | 1190 | 4 | One-hot | 6.72/46.64/46.64 | 80 | 555 | 555 | 0.6978 | 23.86 |
| Squirrel | 2223 | 96219 | 2089 | 5 | Binary Sparse | 47.37/32.30/20.33 | 1053 | 718 | 452 | 0.2072 | 43.28 |
| Wiki | 2405 | 25597 | 4973 | 17 | Unsigned Dense | 14.14/42.91/42.95 | 340 | 1032 | 1033 | 0.6097 | 10.64 |
| Cora | 2708 | 13264 | 1433 | 7 | Sparse | 5.17/18.46/36.93 | 140 | 500 | 1000 | 0.8100 | 4.9 |
| Citeseer | 3327 | 12431 | 3703 | 6 | Sparse | 3.61/15.03/30.06 | 120 | 500 | 1000 | 0.7355 | 3.74 |
| BlogCatalog | 5196 | 348682 | 8189 | 6 | Binary Sparse | 2.31/48.85/48.85 | 120 | 2538 | 2538 | 0.4011 | 67.11 |
| Actor | 7600 | 60918 | 932 | 5 | Binary Sparse | 48.00/32.00/20.00 | 3648 | 2432 | 1520 | 0.2167 | 8.02 |
| AmzPhoto | 7650 | 245812 | 745 | 8 | Binary Dense | 2.09/48.95/48.95 | 160 | 3745 | 3745 | 0.8272 | 32.13 |
| Minesweeper | 10000 | 88804 | 7 | 2 | One-hot | 50.00/25.00/25.00 | 5000 | 2500 | 2500 | 0.6828 | 8.88 |
| WikiCS | 11701 | 442907 | 300 | 10 | Signed Dense | 4.96/15.12/49.97 | 580 | 1769 | 5847 | 0.6543 | 37.85 |
| Tolokers | 11758 | 1049758 | 10 | 2 | Unsigned Dense | 50.00/25.00/25.00 | 5879 | 2939 | 2940 | 0.5945 | 89.28 |
| AmzComputer | 13752 | 505474 | 767 | 10 | Binary Dense | 1.45/49.27/49.27 | 200 | 6776 | 6776 | 0.7772 | 36.76 |
| DBLP | 17716 | 123450 | 1639 | 4 | Binary Sparse | 0.45/49.77/49.77 | 80 | 8818 | 8818 | 0.8279 | 6.97 |
| CoCS | 18333 | 182121 | 6805 | 15 | Sparse | 1.64/49.18/49.18 | 300 | 9016 | 9017 | 0.8081 | 9.93 |
| Pubmed | 19717 | 108365 | 500 | 3 | Binary Dense | 0.30/2.54/5.07 | 60 | 500 | 1000 | 0.8024 | 5.5 |
| FullCora | 19793 | 146635 | 8710 | 70 | Signed Dense | 7.07/46.46/46.47 | 1400 | 9196 | 9197 | 0.5670 | 7.41 |
| Roman Empire | 22662 | 88516 | 300 | 18 | Signed Dense | 50.00/25.00/25.00 | 11331 | 5665 | 5666 | 0.0469 | 3.91 |
| Amazon Ratings | 24492 | 210592 | 300 | 5 | Signed Dense | 50.00/25.00/25.00 | 12246 | 6123 | 6123 | 0.3804 | 8.6 |
| Deezer | 28281 | 213785 | 128 | 2 | Unsigned Dense | 0.14/49.93/49.93 | 40 | 14120 | 14121 | 0.5251 | 7.56 |
| CoPhysics | 34493 | 530417 | 8415 | 5 | Sparse | 0.29/49.85/49.86 | 100 | 17196 | 17197 | 0.9314 | 15.38 |
| Questions | 48921 | 356001 | 301 | 2 | Unsigned Dense | 50.00/25.00/25.00 | 24460 | 12230 | 12231 | 0.8396 | 7.28 |
| OGBN-Arxiv | 169343 | 2484941 | 128 | 40 | Signed Dense | 53.70/17.60/28.70 | 90941 | 29799 | 48603 | 0.6542 | 14.67 |

## M  FULL EXPERIMENT RESULTS

Table 14:  Performance(%) comparison against its predictors, and GraphAny across 28 datasets. **1st**, 2nd best performance is highlighted. Δ rows report relative improvements over each comparator. RGVT consistently outperforms both predictors and GraphAny variants, demonstrating robust generalization across diverse datasets.

| Method | Actor | AirBrazil | AirEU | AirUS | AmzComp | AmzPhoto | AmzRatings | BlogCatalog |
|---|---|---|---|---|---|---|---|---|
| **#Train Nodes** | 3648 | 80 | 80 | 80 | 200 | 160 | 12246 | 120 |
| Linear | **35.38**$_{\pm0.33}$ | 25.38$_{\pm6.44}$ | 23.87$_{\pm2.36}$ | 26.16$_{\pm1.27}$ | 70.42$_{\pm0.45}$ | 77.44$_{\pm0.72}$ | 37.99$_{\pm0.10}$ | 70.80$_{\pm0.49}$ |
| MLP | 35.32$_{\pm0.38}$ | 25.38$_{\pm4.39}$ | 26.62$_{\pm4.11}$ | 25.37$_{\pm1.22}$ | 69.73$_{\pm2.40}$ | 78.64$_{\pm0.63}$ | 45.37$_{\pm0.48}$ | 75.60$_{\pm0.59}$ |
| GraphAny (Wiscon) | 30.68$_{\pm2.06}$ | 42.31$_{\pm0.00}$ | 43.00$_{\pm1.28}$ | 45.16$_{\pm0.21}$ | 81.31$_{\pm0.35}$ | 91.50$_{\pm0.14}$ | 42.25$_{\pm0.61}$ | 77.49$_{\pm1.29}$ |
| GraphAny (Cora) | 29.28$_{\pm1.62}$ | 45.38$_{\pm1.72}$ | 41.87$_{\pm1.59}$ | 45.52$_{\pm0.27}$ | 82.42$_{\pm0.11}$ | 91.10$_{\pm0.11}$ | 42.59$_{\pm0.38}$ | 73.48$_{\pm0.08}$ |
| GraphAny (Arxiv) | 29.70$_{\pm1.11}$ | 42.31$_{\pm0.00}$ | 43.12$_{\pm0.76}$ | 46.31$_{\pm0.49}$ | 82.17$_{\pm0.10}$ | 91.83$_{\pm0.05}$ | 42.51$_{\pm0.19}$ | 75.18$_{\pm1.19}$ |
| GraphAny (Best) | 30.68$_{\pm2.06}$ | 45.38$_{\pm1.72}$ | 43.12$_{\pm0.76}$ | 46.31$_{\pm0.49}$ | 82.42$_{\pm0.11}$ | 91.83$_{\pm0.05}$ | 42.59$_{\pm0.38}$ | 77.49$_{\pm1.29}$ |
| **RGVT**+ Linear (Arxiv) | 33.87$_{\pm0.50}$ | 52.31$_{\pm6.99}$ | 46.25$_{\pm1.93}$ | 57.08$_{\pm3.38}$ | 84.30$_{\pm0.23}$ | 92.02$_{\pm0.30}$ | 42.98$_{\pm0.13}$ | 80.75$_{\pm3.08}$ |
| Δ Linear | ↓4.27% | ↑106.11% | ↑93.76% | ↑118.20% | ↑19.71% | ↑18.83% | ↑13.14% | ↑14.05% |
| Δ GraphAny (Best) | ↑10.40% | ↑15.27% | ↑7.26% | ↑23.26% | ↑2.28% | ↑0.21% | ↑0.92% | ↑4.21% |
| **RGVT**+ MLP (Arxiv) | 34.00$_{\pm1.14}$ | 66.15$_{\pm10.67}$ | 47.00$_{\pm2.70}$ | 56.79$_{\pm2.26}$ | 83.72$_{\pm0.46}$ | 92.15$_{\pm0.36}$ | 48.61$_{\pm1.53}$ | 84.61$_{\pm1.59}$ |
| Δ MLP | ↓3.74% | ↑160.64% | ↑76.56% | ↑123.85% | ↑20.06% | ↑17.18% | ↑7.14% | ↑11.92% |
| Δ GraphAny (Best) | ↑10.82% | ↑45.77% | ↑9.00% | ↑22.63% | ↑1.58% | ↑0.35% | ↑14.13% | ↑9.19% |

| Method | Chameleon | Citeseer | CoCS | CoPhysics | Cora | Cornell | DBLP | Deezer |
|---|---|---|---|---|---|---|---|---|
| **#Train Nodes** | 409 | 120 | 300 | 100 | 140 | 87 | 80 | 40 |
| Linear | 36.39$_{\pm1.95}$ | 49.22$_{\pm0.28}$ | 85.60$_{\pm0.13}$ | 82.21$_{\pm0.10}$ | 48.60$_{\pm1.76}$ | **74.05**$_{\pm1.48}$ | 52.03$_{\pm0.31}$ | 56.39$_{\pm1.30}$ |
| MLP | 36.08$_{\pm4.30}$ | 49.74$_{\pm0.94}$ | 87.23$_{\pm0.53}$ | 86.54$_{\pm0.89}$ | 52.56$_{\pm0.72}$ | 71.89$_{\pm3.08}$ | 53.23$_{\pm0.32}$ | 55.40$_{\pm1.15}$ |
| GraphAny (Wiscon) | 31.14$_{\pm6.89}$ | 68.20$_{\pm0.14}$ | 89.42$_{\pm0.10}$ | 91.92$_{\pm0.12}$ | 76.90$_{\pm0.45}$ | 65.95$_{\pm2.42}$ | 70.59$_{\pm0.82}$ | 54.08$_{\pm0.08}$ |
| GraphAny (Cora) | 30.87$_{\pm6.28}$ | 68.12$_{\pm0.29}$ | 88.77$_{\pm0.03}$ | 92.25$_{\pm0.02}$ | 76.84$_{\pm0.05}$ | 63.24$_{\pm2.42}$ | 72.18$_{\pm0.05}$ | 53.98$_{\pm0.04}$ |
| GraphAny (Arxiv) | 30.41$_{\pm6.12}$ | 67.82$_{\pm0.54}$ | 89.00$_{\pm0.17}$ | 92.20$_{\pm0.02}$ | 77.70$_{\pm0.20}$ | 64.86$_{\pm0.00}$ | 71.47$_{\pm0.18}$ | 54.20$_{\pm0.13}$ |
| GraphAny (Best) | 31.14$_{\pm6.89}$ | 68.20$_{\pm0.14}$ | 89.42$_{\pm0.10}$ | 92.25$_{\pm0.02}$ | 77.70$_{\pm0.20}$ | 65.95$_{\pm2.42}$ | 72.18$_{\pm0.05}$ | 54.20$_{\pm0.13}$ |
| **RGVT**+ Linear (Arxiv) | 41.34$_{\pm0.99}$ | 71.96$_{\pm0.38}$ | 91.32$_{\pm0.42}$ | 92.71$_{\pm0.35}$ | 81.32$_{\pm0.64}$ | 73.51$_{\pm4.44}$ | 81.29$_{\pm1.01}$ | 57.79$_{\pm0.29}$ |
| Δ Linear | ↑13.60% | ↑46.20% | ↑6.68% | ↑12.77% | ↑67.33% | ↓0.73% | ↑56.24% | ↑2.48% |
| Δ GraphAny (Best) | ↑32.76% | ↑5.51% | ↑2.12% | ↑0.50% | ↑4.66% | ↑11.46% | ↑12.62% | ↑6.62% |
| **RGVT**+ MLP (Arxiv) | 44.74$_{\pm2.17}$ | 70.02$_{\pm1.34}$ | 92.10$_{\pm0.14}$ | 92.62$_{\pm0.60}$ | 81.18$_{\pm1.15}$ | 74.05$_{\pm1.48}$ | 81.08$_{\pm1.22}$ | 57.58$_{\pm0.61}$ |
| Δ MLP | ↑24.00% | ↑40.77% | ↑5.58% | ↑7.03% | ↑54.45% | ↑3.00% | ↑52.32% | ↑3.94% |
| Δ GraphAny (Best) | ↑43.67% | ↑2.67% | ↑3.00% | ↑0.40% | ↑4.48% | ↑12.28% | ↑12.33% | ↑6.24% |

| Method | Minesweeper | Pubmed | Questions | Roman | Squirrel | Texas | Tolokers | Wiki |
|---|---|---|---|---|---|---|---|---|
| **#Train Nodes** | 5000 | 60 | 24460 | 11331 | 1053 | 87 | 5879 | 340 |
| Linear | 80.00$_{\pm0.00}$ | 68.68$_{\pm0.90}$ | 97.05$_{\pm0.00}$ | 63.19$_{\pm0.04}$ | 30.62$_{\pm0.51}$ | 78.38$_{\pm0.00}$ | 78.12$_{\pm0.11}$ | 71.13$_{\pm1.14}$ |
| MLP | 80.00$_{\pm0.00}$ | 70.28$_{\pm1.00}$ | 97.12$_{\pm0.09}$ | 64.43$_{\pm0.28}$ | 34.73$_{\pm2.40}$ | **79.46**$_{\pm1.48}$ | 78.28$_{\pm0.05}$ | 72.62$_{\pm1.43}$ |
| GraphAny (Wiscon) | 80.26$_{\pm0.12}$ | 77.50$_{\pm0.23}$ | 97.11$_{\pm0.01}$ | 63.88$_{\pm0.20}$ | 24.97$_{\pm0.82}$ | 74.89$_{\pm2.63}$ | 78.19$_{\pm0.02}$ | 57.75$_{\pm0.46}$ |
| GraphAny (Cora) | **80.42**$_{\pm0.13}$ | 76.52$_{\pm0.04}$ | 97.08$_{\pm0.01}$ | 63.59$_{\pm0.62}$ | 24.60$_{\pm0.86}$ | 71.56$_{\pm1.93}$ | 78.17$_{\pm0.06}$ | 57.81$_{\pm0.10}$ |
| GraphAny (Arxiv) | 80.34$_{\pm0.15}$ | 76.68$_{\pm0.08}$ | 97.10$_{\pm0.01}$ | 63.74$_{\pm0.62}$ | 26.01$_{\pm0.44}$ | 71.52$_{\pm5.78}$ | 78.14$_{\pm0.03}$ | 61.06$_{\pm0.42}$ |
| GraphAny (Best) | **80.42**$_{\pm0.13}$ | 77.50$_{\pm0.23}$ | 97.11$_{\pm0.01}$ | 63.88$_{\pm0.20}$ | 26.01$_{\pm0.44}$ | 74.89$_{\pm2.63}$ | 78.19$_{\pm0.02}$ | 61.06$_{\pm0.42}$ |
| **RGVT**+ Linear (Arxiv) | 79.77$_{\pm0.27}$ | **79.02**$_{\pm1.10}$ | **97.13**$_{\pm0.02}$ | 67.71$_{\pm0.87}$ | 39.38$_{\pm0.81}$ | 75.14$_{\pm2.96}$ | 78.10$_{\pm0.29}$ | 72.76$_{\pm0.98}$ |
| Δ Linear | ↓0.29% | ↑15.06% | ↑0.08% | ↑7.15% | ↑28.61% | ↓4.13% | ↓0.03% | ↑2.29% |
| Δ GraphAny (Best) | ↓0.81% | ↑1.96% | ↑0.02% | ↑6.00% | ↑51.40% | ↑0.33% | ↓0.12% | ↑19.16% |
| **RGVT**+ MLP (Arxiv) | 79.98$_{\pm0.18}$ | 78.72$_{\pm0.70}$ | **97.13**$_{\pm0.02}$ | 70.18$_{\pm0.28}$ | 41.77$_{\pm2.06}$ | 78.92$_{\pm1.21}$ | 79.23$_{\pm0.42}$ | 74.54$_{\pm0.76}$ |
| Δ MLP | ↓0.02% | ↑12.01% | ↑0.01% | ↑8.92% | ↑20.27% | ↓0.68% | ↑1.21% | ↑2.64% |
| Δ GraphAny (Best) | ↓0.55% | ↑1.57% | ↑0.02% | ↑9.86% | ↑60.59% | ↑5.38% | ↑1.33% | ↑22.08% |

| Method | Wisconsin | WikiCS | OGBN-Arxiv | FullCora | **Total Avg.** |
|---|---|---|---|---|---|
| **#Train Nodes** | 120 | 580 | 90941 | 1400 | - |
| Linear | **79.22**$_{\pm1.07}$ | 72.50$_{\pm0.04}$ | 52.44$_{\pm0.04}$ | 40.32$_{\pm0.20}$ | 59.41$_{\pm0.84}$ |
| MLP | 74.90$_{\pm2.15}$ | 72.08$_{\pm0.19}$ | 53.80$_{\pm0.14}$ | 39.75$_{\pm0.44}$ | 60.43$_{\pm1.20}$ |
| GraphAny (Wiscon) | 66.28$_{\pm4.02}$ | 74.61$_{\pm0.80}$ | 57.77$_{\pm0.45}$ | 57.12$_{\pm0.21}$ | 64.72$_{\pm0.96}$ |
| GraphAny (Cora) | 61.96$_{\pm6.30}$ | 74.91$_{\pm0.92}$ | 58.58$_{\pm0.10}$ | 57.21$_{\pm0.09}$ | 64.30$_{\pm0.94}$ |
| GraphAny (Arxiv) | 64.31$_{\pm5.08}$ | 75.53$_{\pm0.78}$ | 58.63$_{\pm0.14}$ | 58.08$_{\pm0.08}$ | 64.71$_{\pm0.89}$ |
| GraphAny (Best) | 66.28$_{\pm4.02}$ | 75.53$_{\pm0.78}$ | 58.63$_{\pm0.14}$ | 58.08$_{\pm0.08}$ | 65.30$_{\pm0.93}$ |
| **RGVT**+ Linear (Arxiv) | 76.86$_{\pm1.64}$ | **79.59**$_{\pm0.16}$ | 70.14$_{\pm0.28}$ | **64.35**$_{\pm0.77}$ | 70.03$_{\pm1.26}$ |
| Δ Linear | ↓2.98% | ↑9.78% | ↑33.75% | ↑59.60% | ↑17.88% |
| Δ GraphAny (Best) | ↑15.96% | ↑5.38% | ↑19.63% | ↑10.80% | ↑7.24% |
| **RGVT**+ MLP (Arxiv) | 71.76$_{\pm1.75}$ | **79.59**$_{\pm0.19}$ | 71.11$_{\pm0.28}$ | 62.35$_{\pm2.22}$ | 71.13$_{\pm1.41}$ |
| Δ MLP | ↓4.19% | ↑10.42% | ↑32.17% | ↑56.86% | ↑17.71% |
| Δ GraphAny (Best) | ↑8.27% | ↑5.38% | ↑21.29% | ↑7.35% | ↑8.93% |

Table 15: Performance (%) comparison with 12 individually trained GNN models across 28 datasets. 1st, 2nd, 3rd best results are highlighted. RGVT achieves the best performance on average, ranking 1st with the MLP predictor and 2nd with the linear predictor.

| Method | Actor | AirBrazil | AirEU | AirUSA | AmzComp | AmzPhoto | AmzRatings | BlogCatalog |
|---|---|---|---|---|---|---|---|---|
| #Train Nodes | 3648 | 80 | 80 | 80 | 200 | 160 | 12246 | 120 |
| GIN | $27.75_{\pm1.52}$ | $49.23_{\pm10.67}$ | $40.50_{\pm3.11}$ | $51.32_{\pm1.47}$ | $36.26_{\pm1.10}$ | $28.02_{\pm6.08}$ | $46.01_{\pm2.73}$ | $17.00_{\pm0.51}$ |
| GAT | $28.57_{\pm0.82}$ | $44.62_{\pm14.54}$ | $40.00_{\pm1.93}$ | $51.24_{\pm2.90}$ | $84.43_{\pm1.28}$ | $89.40_{\pm0.71}$ | $48.99_{\pm0.56}$ | $55.16_{\pm4.77}$ |
| GATv2 | $29.80_{\pm0.46}$ | $54.62_{\pm8.34}$ | $40.12_{\pm2.23}$ | $47.14_{\pm2.30}$ | $84.51_{\pm0.71}$ | $89.54_{\pm0.39}$ | $49.57_{\pm0.67}$ | $57.94_{\pm3.22}$ |
| S²GC | $28.93_{\pm0.21}$ | $54.62_{\pm3.22}$ | $41.12_{\pm1.20}$ | $50.92_{\pm0.49}$ | $84.70_{\pm0.10}$ | $92.07_{\pm0.27}$ | $42.05_{\pm0.20}$ | $80.09_{\pm0.06}$ |
| SGC | $29.70_{\pm0.08}$ | $71.54_{\pm10.39}$ | $42.62_{\pm0.81}$ | $54.49_{\pm1.45}$ | $84.39_{\pm0.09}$ | $92.26_{\pm0.16}$ | $43.07_{\pm0.08}$ | $71.06_{\pm0.04}$ |
| JKNet | $30.53_{\pm0.36}$ | $56.92_{\pm15.72}$ | $41.88_{\pm0.99}$ | $53.26_{\pm2.71}$ | $84.96_{\pm0.30}$ | $91.41_{\pm0.60}$ | $48.93_{\pm0.40}$ | $72.11_{\pm0.31}$ |
| APPNP | $31.72_{\pm0.54}$ | $36.92_{\pm3.44}$ | $31.25_{\pm0.99}$ | $47.50_{\pm0.52}$ | $84.39_{\pm0.18}$ | $91.87_{\pm0.21}$ | $50.63_{\pm0.44}$ | $86.00_{\pm0.52}$ |
| GCN | $30.37_{\pm0.44}$ | $63.08_{\pm3.44}$ | $41.50_{\pm0.84}$ | $53.30_{\pm1.18}$ | $84.69_{\pm0.18}$ | $91.71_{\pm0.18}$ | $49.17_{\pm0.43}$ | $71.75_{\pm0.24}$ |
| GCNII | $34.22_{\pm0.35}$ | $46.15_{\pm5.44}$ | $41.00_{\pm9.45}$ | $51.24_{\pm1.69}$ | $84.81_{\pm0.23}$ | $91.39_{\pm0.30}$ | $50.37_{\pm0.17}$ | $87.43_{\pm0.69}$ |
| SAGE | $36.37_{\pm0.93}$ | $34.62_{\pm7.20}$ | $37.50_{\pm4.35}$ | $48.18_{\pm1.72}$ | $83.65_{\pm0.19}$ | $90.74_{\pm0.42}$ | $50.86_{\pm0.41}$ | $80.27_{\pm3.55}$ |
| GPRGNN | $34.51_{\pm0.31}$ | $31.54_{\pm4.21}$ | $34.37_{\pm1.65}$ | $51.57_{\pm0.59}$ | $85.55_{\pm0.31}$ | $92.81_{\pm0.18}$ | $48.74_{\pm0.28}$ | $88.34_{\pm0.27}$ |
| UniMP | $35.97_{\pm1.39}$ | $44.62_{\pm9.65}$ | $39.25_{\pm3.14}$ | $47.64_{\pm1.46}$ | $84.06_{\pm0.70}$ | $92.15_{\pm0.25}$ | $51.30_{\pm0.77}$ | $80.70_{\pm3.24}$ |
| **RGVT** + Linear | $33.87_{\pm0.50}$ | $52.31_{\pm6.99}$ | $46.25_{\pm1.93}$ | $57.08_{\pm3.38}$ | $84.30_{\pm0.23}$ | $92.02_{\pm0.30}$ | $42.98_{\pm0.13}$ | $80.75_{\pm3.08}$ |
| **RGVT** + MLP | $34.00_{\pm1.14}$ | $66.15_{\pm10.67}$ | $47.00_{\pm2.70}$ | $56.79_{\pm2.26}$ | $83.72_{\pm0.46}$ | $92.15_{\pm0.36}$ | $48.61_{\pm1.53}$ | $84.61_{\pm1.59}$ |

| Method | Chameleon | Citeseer | CoCS | CoPhysics | Cora | Cornell | DBLP | Deezer |
|---|---|---|---|---|---|---|---|---|
| #Train Nodes | 409 | 120 | 300 | 100 | 140 | 87 | 80 | 40 |
| GIN | $35.15_{\pm3.14}$ | $47.48_{\pm10.03}$ | $84.20_{\pm0.79}$ | $85.98_{\pm4.54}$ | $70.06_{\pm3.28}$ | $37.84_{\pm2.70}$ | $71.68_{\pm1.56}$ | $55.33_{\pm0.51}$ |
| GAT | $39.18_{\pm2.70}$ | $69.46_{\pm1.36}$ | $90.27_{\pm0.57}$ | $91.22_{\pm2.06}$ | $79.06_{\pm1.04}$ | $40.00_{\pm2.26}$ | $79.07_{\pm0.78}$ | $55.57_{\pm0.34}$ |
| GATv2 | $38.76_{\pm0.92}$ | $68.60_{\pm0.62}$ | $88.71_{\pm0.98}$ | $89.43_{\pm3.15}$ | $79.98_{\pm1.02}$ | $37.84_{\pm3.82}$ | $78.28_{\pm1.92}$ | $55.69_{\pm0.40}$ |
| S²GC | $41.75_{\pm0.82}$ | $67.32_{\pm0.37}$ | $90.75_{\pm0.03}$ | $91.69_{\pm0.04}$ | $78.40_{\pm0.71}$ | $43.78_{\pm2.26}$ | $70.27_{\pm0.07}$ | $54.48_{\pm0.36}$ |
| SGC | $43.09_{\pm0.28}$ | $67.98_{\pm0.58}$ | $90.28_{\pm0.03}$ | $92.70_{\pm0.03}$ | $78.46_{\pm0.83}$ | $43.78_{\pm1.21}$ | $74.40_{\pm0.05}$ | $54.56_{\pm0.52}$ |
| JKNet | $41.75_{\pm1.09}$ | $68.32_{\pm1.68}$ | $90.28_{\pm0.22}$ | $92.65_{\pm0.15}$ | $81.96_{\pm0.77}$ | $40.54_{\pm0.00}$ | $75.96_{\pm1.32}$ | $56.30_{\pm0.43}$ |
| APPNP | $40.10_{\pm0.43}$ | $68.68_{\pm0.43}$ | $91.13_{\pm0.09}$ | $93.45_{\pm0.07}$ | $81.38_{\pm0.95}$ | $47.03_{\pm6.22}$ | $77.75_{\pm0.28}$ | $55.97_{\pm0.53}$ |
| GCN | $39.90_{\pm1.19}$ | $69.22_{\pm0.62}$ | $90.60_{\pm0.09}$ | $92.85_{\pm0.24}$ | $81.26_{\pm0.77}$ | $45.41_{\pm3.52}$ | $79.52_{\pm1.03}$ | $57.09_{\pm0.62}$ |
| GCNII | $41.55_{\pm2.54}$ | $63.76_{\pm0.92}$ | $91.07_{\pm0.10}$ | $92.76_{\pm0.25}$ | $76.42_{\pm2.62}$ | $37.84_{\pm2.70}$ | $78.44_{\pm1.62}$ | $55.82_{\pm0.43}$ |
| SAGE | $40.31_{\pm1.43}$ | $66.96_{\pm2.24}$ | $90.55_{\pm0.09}$ | $92.10_{\pm0.54}$ | $78.96_{\pm0.72}$ | $67.03_{\pm3.52}$ | $73.47_{\pm0.37}$ | $56.20_{\pm1.07}$ |
| GPRGNN | $40.93_{\pm1.24}$ | $70.30_{\pm0.62}$ | $91.83_{\pm0.21}$ | $93.20_{\pm0.10}$ | $82.08_{\pm0.66}$ | $68.65_{\pm3.63}$ | $78.41_{\pm0.15}$ | $56.89_{\pm0.26}$ |
| UniMP | $41.34_{\pm3.16}$ | $68.56_{\pm0.19}$ | $89.84_{\pm0.39}$ | $91.53_{\pm0.76}$ | $79.18_{\pm1.13}$ | $62.16_{\pm4.27}$ | $73.44_{\pm0.39}$ | $56.32_{\pm0.88}$ |
| **RGVT** + Linear | $41.34_{\pm0.99}$ | $71.96_{\pm0.38}$ | $91.32_{\pm0.42}$ | $92.71_{\pm0.35}$ | $81.32_{\pm0.64}$ | $73.51_{\pm4.44}$ | $81.29_{\pm1.01}$ | $57.79_{\pm0.29}$ |
| **RGVT** + MLP | $44.74_{\pm2.17}$ | $70.02_{\pm1.34}$ | $92.10_{\pm0.14}$ | $92.62_{\pm0.60}$ | $81.18_{\pm1.15}$ | $74.05_{\pm1.48}$ | $81.08_{\pm1.22}$ | $57.58_{\pm0.61}$ |

| Method | Minesweeper | Pubmed | Questions | Roman | Squirrel | Texas | Tolokers | Wiki |
|---|---|---|---|---|---|---|---|---|
| #Train Nodes | 5000 | 60 | 24460 | 11331 | 1053 | 87 | 5879 | 340 |
| GIN | $80.18_{\pm0.26}$ | $70.22_{\pm6.97}$ | $97.02_{\pm0.01}$ | $51.41_{\pm1.83}$ | $34.73_{\pm0.00}$ | $68.11_{\pm5.20}$ | $78.16_{\pm0.02}$ | $41.28_{\pm15.08}$ |
| GAT | $81.70_{\pm0.53}$ | $76.26_{\pm0.70}$ | $97.06_{\pm0.02}$ | $43.42_{\pm0.18}$ | $36.46_{\pm0.50}$ | $63.78_{\pm6.22}$ | $81.67_{\pm0.18}$ | $58.64_{\pm1.93}$ |
| GATv2 | $82.60_{\pm0.78}$ | $75.56_{\pm1.28}$ | $97.04_{\pm0.02}$ | $59.97_{\pm0.82}$ | $38.41_{\pm2.13}$ | $62.16_{\pm6.34}$ | $80.73_{\pm1.55}$ | $54.52_{\pm2.90}$ |
| S²GC | $81.24_{\pm0.50}$ | $76.68_{\pm0.04}$ | $97.06_{\pm0.00}$ | $48.73_{\pm0.11}$ | $37.52_{\pm0.25}$ | $62.16_{\pm3.31}$ | $78.65_{\pm0.06}$ | $70.94_{\pm1.00}$ |
| SGC | $81.52_{\pm0.30}$ | $77.54_{\pm0.05}$ | $97.09_{\pm0.00}$ | $43.71_{\pm0.06}$ | $38.14_{\pm0.20}$ | $55.68_{\pm1.48}$ | $78.58_{\pm0.04}$ | $70.47_{\pm0.45}$ |
| JKNet | $81.79_{\pm0.31}$ | $75.70_{\pm0.57}$ | $97.06_{\pm0.03}$ | $53.76_{\pm4.05}$ | $34.34_{\pm1.83}$ | $63.24_{\pm6.22}$ | $80.97_{\pm0.28}$ | $70.15_{\pm0.94}$ |
| APPNP | $81.47_{\pm0.11}$ | $76.08_{\pm0.24}$ | $97.08_{\pm0.02}$ | $60.40_{\pm0.31}$ | $35.44_{\pm0.77}$ | $67.57_{\pm1.91}$ | $78.14_{\pm0.04}$ | $73.90_{\pm0.85}$ |
| GCN | $80.16_{\pm0.26}$ | $76.90_{\pm0.42}$ | $97.06_{\pm0.02}$ | $45.33_{\pm0.25}$ | $38.54_{\pm1.64}$ | $65.41_{\pm2.26}$ | $80.60_{\pm0.12}$ | $69.24_{\pm0.76}$ |
| GCNII | $86.04_{\pm0.41}$ | $74.76_{\pm2.91}$ | $97.18_{\pm0.06}$ | $75.80_{\pm0.41}$ | $35.97_{\pm0.58}$ | $62.16_{\pm2.70}$ | $81.79_{\pm0.66}$ | $75.35_{\pm0.66}$ |
| SAGE | $87.06_{\pm0.30}$ | $76.68_{\pm0.91}$ | $97.14_{\pm0.04}$ | $77.69_{\pm0.48}$ | $38.19_{\pm1.12}$ | $82.16_{\pm6.51}$ | $80.66_{\pm0.29}$ | $72.53_{\pm1.22}$ |
| GPRGNN | $84.10_{\pm0.15}$ | $77.44_{\pm0.44}$ | $97.21_{\pm0.02}$ | $74.24_{\pm0.30}$ | $35.80_{\pm0.96}$ | $73.51_{\pm6.16}$ | $78.41_{\pm0.06}$ | $74.60_{\pm0.37}$ |
| UniMP | $88.25_{\pm0.17}$ | $76.46_{\pm0.65}$ | $97.17_{\pm0.03}$ | $80.31_{\pm0.56}$ | $40.18_{\pm1.97}$ | $73.51_{\pm3.52}$ | $81.46_{\pm0.38}$ | $72.93_{\pm1.12}$ |
| **RGVT** + Linear | $79.77_{\pm0.27}$ | $79.02_{\pm1.10}$ | $97.13_{\pm0.02}$ | $67.71_{\pm0.87}$ | $39.38_{\pm0.81}$ | $75.14_{\pm2.96}$ | $78.10_{\pm0.29}$ | $72.76_{\pm0.98}$ |
| **RGVT** + MLP | $79.98_{\pm0.18}$ | $78.72_{\pm0.70}$ | $97.13_{\pm0.02}$ | $70.18_{\pm0.28}$ | $41.77_{\pm2.06}$ | $78.92_{\pm1.21}$ | $79.23_{\pm0.42}$ | $74.54_{\pm0.76}$ |

| Method | Wisconsin | WikiCS | OGBN-Arxiv | FullCora | Total Avg. |
|---|---|---|---|---|---|
| #Train Nodes | 120 | 580 | 90941 | 1400 | - |
| GIN | $54.12_{\pm5.30}$ | $56.85_{\pm13.75}$ | $65.77_{\pm1.08}$ | $57.80_{\pm0.32}$ | $54.98_{\pm3.70}$ |
| GAT | $51.37_{\pm2.56}$ | $79.44_{\pm0.40}$ | $71.89_{\pm0.24}$ | $60.71_{\pm0.16}$ | $63.88_{\pm1.87}$ |
| GATv2 | $53.73_{\pm3.56}$ | $79.78_{\pm0.53}$ | $72.13_{\pm0.11}$ | $60.80_{\pm0.12}$ | $64.57_{\pm1.83}$ |
| S²GC | $53.73_{\pm2.24}$ | $79.90_{\pm0.05}$ | $69.17_{\pm0.02}$ | $59.84_{\pm0.03}$ | $65.31_{\pm0.64}$ |
| SGC | $53.33_{\pm3.51}$ | $79.40_{\pm0.06}$ | $68.69_{\pm0.03}$ | $59.86_{\pm0.00}$ | $65.66_{\pm0.82}$ |
| JKNet | $55.29_{\pm1.64}$ | $79.46_{\pm0.26}$ | $71.73_{\pm0.24}$ | $61.99_{\pm1.00}$ | $66.19_{\pm1.59}$ |
| APPNP | $56.08_{\pm1.07}$ | $78.80_{\pm0.06}$ | $70.85_{\pm0.15}$ | $63.65_{\pm0.09}$ | $66.26_{\pm0.77}$ |
| GCN | $53.33_{\pm1.64}$ | $79.07_{\pm0.24}$ | $71.19_{\pm0.30}$ | $62.69_{\pm0.13}$ | $66.46_{\pm0.82}$ |
| GCNII | $61.57_{\pm2.63}$ | $79.03_{\pm0.17}$ | $72.05_{\pm0.08}$ | $58.31_{\pm0.44}$ | $67.30_{\pm1.47}$ |
| SAGE | $62.75_{\pm3.10}$ | $78.77_{\pm0.48}$ | $71.22_{\pm0.09}$ | $61.35_{\pm0.09}$ | $68.36_{\pm1.55}$ |
| GPRGNN | $64.71_{\pm2.40}$ | $79.46_{\pm0.31}$ | $69.45_{\pm0.41}$ | $64.31_{\pm0.11}$ | $68.68_{\pm0.94}$ |
| UniMP | $65.88_{\pm9.76}$ | $79.80_{\pm0.13}$ | $71.78_{\pm0.12}$ | $62.26_{\pm0.11}$ | $68.86_{\pm1.80}$ |
| **RGVT** + Linear | $76.86_{\pm1.64}$ | $79.59_{\pm0.16}$ | $70.14_{\pm0.28}$ | $64.35_{\pm0.77}$ | $70.03_{\pm1.26}$ |
| **RGVT** + MLP | $71.76_{\pm1.75}$ | $79.59_{\pm0.19}$ | $71.11_{\pm0.28}$ | $62.35_{\pm2.22}$ | $71.13_{\pm1.41}$ |

