# OpenReview forum: "Unlocking Universal Graph Knowledge in the View Space"
_ICLR.cc/2026/Conference — Submitted to ICLR 2026_

### Official Review · Reviewer_M8Kc · 2025-10-18

**Soundness:** 2
**Presentation:** 3
**Contribution:** 2
**Rating:** 4
**Confidence:** 5

**Summary:**

This paper proposes a new way to train inductive gnn model by projecting the original features into a view space. By learning a shared projecter, the learned representation can be projected to desired task dimension through weight sharing. A recurrent variant is further adopted to deal with tasks requiring different hops.

**Strengths:**

1. the problem is interesting
2. the presentation is generally good
3. the performance boost over graphany is clear, which demonstrates that universal representation is much better than directly aggregating the closed-form solution

**Weaknesses:**

1. I don't agree with the claim in the abstract. Graphany is obviously not the only universal graph model. First, you can't say it's universal. graph is not just about node classification. In their papers, they also don't claim they are graph foundation models (they just say fully-inductive classification). Second, if graphany is universal, they i don't see why others like OneForAll are not universal.
2. In the introduction part, the motivation makes me feel like the authors think the feature heterogeneity is the main problem of building a graph foundation model. I strongly disagree with this viewpoint. FIrst, it's obvious that structure heterogeneity is much harder. For example, from a geometric perspective, the 1-order (entity-level) and 2-order (link-level) tasks are not compatible. Moreover, the homophily-heterophily problem is very challenging to solve. Second, in practice, the feature heterogeneity may be not a "real" problem. There are many relational foundation mdoels like kumorfm and tabular foundation models that can work on heterogeneous feature types. You just need a type-aware encoder, which works generally well.
3. In the introduction part, authors point out that the way OneForAll takes results in clear performance loss. I also need to point out that the way GraphAny leads to even more performance loss by the inductive transformation.
4. Section 3 is in general identical to Graphany.
5. One obvious drawback of recurrent one is the expressivenss problem, You have to weight share the parameters across layers, and non-identical weight learning for heterophilous graph is very important (for example, in EvenNet)
6. the experimental dataset are selected with relatively weak features. Vanilla GNN may be much better with better features like LLM-encoded ones.
7. THis model can only work for node classification tasks, and can't do in-context learning, zero-shot learning, generation. I would say it's far away from a foundation model.

**Questions:**

What's the main design component that makes GVT so much better than graphany? Better explain this in experiment sections.

---

> ### Author Response · Authors · 2025-11-22
>
> We sincerely appreciate your detailed review and insightful questions.
> We first refer you to the common response above, where we clarify our terminology and scope, which we believe contributed to the unintended impression of overstatement. We also summarize the revisions made in the manuscript to address these issues.
>
> ---
>
> **W1. Overclaim due to the term “universal”.**
>
> As noted in the common response, we previously used the term “universal” to imply “fully inductive”; however, we now recognize that this phrasing is inappropriate and have removed it throughout the manuscript.
> In addition, we no longer present our approach as a graph foundation model; instead, we clearly position it as a solution for *fully inductive representation learning* (FI-NRL).
>
> ---
>
> **W2. Task-level and structural heterogeneity pose more fundamental challenges for GFMs. Feature heterogeneity may not be the real problem of GFMs since it can be easily addressed by the type-aware encoder and tabular foundation models.**
>
> We hope that the common response and the revised manuscript address the concerns regarding the impression of overstatement and distinguish our goal from the broader aims of GFMs.
> Below, we further explain our perspective on inductive graph learning, focusing on the respective roles of structural and feature heterogeneity and why our work centers on feature one.
>
> Structural heterogeneity is indeed an important challenge in inductive graph learning, and we fully acknowledge its significance. The community has made substantial progress in this direction: GNN message passing enables inductive inference on graphs with arbitrary numbers of nodes and edges, and many works strengthen structural generalization on unseen topologies through architectural modifications, training strategies, objectives, etc.
> Thus, while structural heterogeneity remains an open problem, it is one which effective tools are already actively being developed.
>
> In contrast, feature heterogeneity in inductive graph learning remains largely underexplored.
> A single graph model capable of performing inference on any dataset, regardless of its feature space, would dramatically expand the model’s applicability, much like how LLMs operate on arbitrary text inputs.
> Despite its importance, this problem remains underexplored because it is inherently difficult.
>
> The core challenge stems from the nature of feature transformations: deep learning architectures---including GNNs and GFMs---mostly learn and operate within a feature space.
> Consequently, a model must have encountered a particular feature space during training to process it at inference time.
> This dependence fundamentally restricts inference to datasets with unseen or incompatible feature types.
> To overcome this limitation, we propose learning and operating in a different space---the view space---which provides a principled and effective way to achieve fully inductive learning.
>
> **About Graph Foundation Models**
> While graph foundation models (GFMs) and relational foundation models (RFMs), including those equipped with type-aware encoders can support multiple feature types, they are not fully inductive.
> These models require prior exposure to the same or similar feature types, or they depend on per-dataset fine-tuning.
> In contrast, a fully inductive model must operate on *any* feature space, including formats that were never observed during training.
> This fundamentally requires *type-agnostic*, rather than type-aware processing.
> This inherent reliance on observed feature types limits the applicability of GFMs and RFMs when encountering unseen or incompatible feature spaces. Consequently, they are not solution of fully inductive learning problem.
>
>
> **About Tabular Foundation Models**
> Tabular foundation models, particularly prior-data fitted networks (PFNs) [1], can accommodate heterogeneous feature spaces and operate in a fully inductive manner. Several recent ICLR submissions have therefore adapted these models for graph data [2, 3].
> However, PFNs and their variants produce predictions directly through sample-space operations rather than learning reusable representations.
> Since our goal is representation learning, these methods are also not a solution for FI-NRL.
>
> [1] Accurate predictions on small data with a tabular foundation model (Nature 2025)\
> [2] Bringing Graphs to the Table: Zero-Shot Node Classification via Tabular Foundation Models (ICLR 2026 submission)\
> [3] GraphPFN: A Prior-Data Fitted Graph Foundation Model (ICLR 2026 submission)

---

> ### Author Response · Authors · 2025-11-22
>
> **W3. GraphAny leads to even more performance loss by the inductive transformation.**
>
> We agree that a fully inductive framework may exhibit performance degradation when compared with transductive GFMs, as it cannot exploit dataset- or domain-specific information.
> As clarified in our common response, we treat the fully inductive setting as a distinct problem.
>
> ---
>
> **W4. Section 3 is in general identical to GraphAny.**
>
> The key distinction is that GraphAny contends feature-permutation-*invariance*, whereas we propose feature-permutation-*equivariance* as a condition for achieving a fully inductive model.
>
> Permutation equivariance is a stronger and more demanding condition.
> First, any invariant function is an invariant readout of an underlying equivariant representation, so an equivariant function offers strictly greater expressivity.
> Second, invariant mappings discard feature-wise structures, such as cross-channel interactions, limiting the information available on representations.
> Thus, our proposal of feature-permutation-equivariance is not only a different choice but also a core contribution to FI-NRL.
> We have included this explanation, along with theoretical proofs, in the revised manuscript (lines 143–149).
>
> ---
>
> **W5. One obvious drawback of recurrent one is the expressiveness problem, and non-identical weight learning for heterophilous graph is very important.**
>
> Thank you for the insightful question. We agree that expressiveness is an important point, particularly for heterohpilous graphs.
>
> Although RGVT shares weights across layers, its node–feature dynamic aggregation mechanism (Section 5) ensures that each layer behaves differently.
> The aggregation is guided by the view vector at each layer, which you can think of as model decides how to aggregate based on a simulation of aggregated signals.
> Because the view vector changes for each node–feature pair at every layer, the resulting aggregation also differs from layer to layer.
>
> Furthermore, for an example of EvenNet, GVT can also suppress odd-hop contributions.
> It can do so either by assigning negligible weights to the odd-hop view-finder outputs or by learning combinations of multiple views that diminish the effect of odd-hop signals.
>
> In addition, GVT's node-feature dynamic aggregation mechanism allows even each feature channel to adopt its own aggregation rule.
> Recent work [5] shows that different feature channels often favor different forms of aggregation, and that separating these behaviors across architectures improves performance on heterophilous graphs.
> GVT generalizes this principle: its channel-wise dynamic aggregation adaptively chooses how each feature should be aggregated at each layer.
>
> These properties are also reflected in our experiment results.
> Our main experiments are conducted on 28 different graphs for downstream evaluation, while 11 of them exhibit heterophily, with homophily ratios below 0.5.
> RGVT does not show noticeable performance degradation on these datasets compared with homophilous datasets.
> The same trend appears in our extended experiments on graphs with LLM-embedded features (described later), indicating that RGVT demonstrates strong structural generalization on heterophilous graphs.
>
> [4] Let Your Features Tell The Differences: Understanding Graph Convolution By Feature Splitting (ICLR 2025)
>
> ---
>
> **W6. Vanilla GNN may be much better with rich features like LLM-encoded ones.**
>
> Thank you for this insightful comment.
> We conduct additional experiments using graphs with LLM-embedded features constructed in recent work [5] (Appendix G).
> We evaluate on seven datasets and compare RGVT and GraphAny with three GNNs (GCN, UniMP, and GPRGNN) that demonstrated strong performance in our main experiments.
>
> As you anticipated, unlike the main experiments, GraphAny exhibited substantial performance degradation relative to these GNN baselines. In contrast, our pretrained RGVT, combined with an MLP predictor, achieved competitive results and ranked first on two datasets.
>
> These results reinforce the strength of RGVT’s feature-permutation-equivariant design, which effectively preserves feature-wise structure, as noted in our response to W4.
> Notably, even the rich feature relationships encoded by LLM-generated embeddings are well maintained in the learned representations and can be effectively exploited by the MLP predictor.
> This demonstrates that RGVT remains a robust solution for FI-NRL regardless of the richness of the input feature space.
>
> [5] Generalization principles for inference over text-attributed graphs with large language models (ICLR 2025)

---

> ### Author Response · Authors · 2025-11-22
>
> **W7. This model can only work for node classification tasks, and can't do in-context learning, zero-shot learning, generation. I would say it's far away from a foundation model.**
>
> We apologize again for the confusion.
> As clarified in the common response, our goal is fully inductive *representation learning*, not building an end-to-end foundation model.
> Although this lies outside our primary scope, one natural extension to in-context learning would be to use a tabular foundation model [1] by feeding the representations produced by our method.
>
> ---
>
> **Q1. What's the main design component that makes GVT so much better than graphany? Better explain this in experiment sections.**
>
> Thank you for thoughtful question. We have reflected your suggestion in the revised manuscript. In particular, we add a dedicated paragraph in Section 7 (Related Works) to clearly distinguish our work from GraphAny (lines 357–365) and provide additional explanation in Section 8 (Experiments, lines 426–431).
>
> First, GVT adopts a *representation-learning* framework rather than GraphAny’s prediction-attention mechanism.
> This yields far more expressive node embeddings and decouples representation quality from the predictor, allowing stronger non-linear models such as MLPs to be used as a predictor.
>
> Second, GVT is *feature-permutation-equivariant* that preserves feature-wise structure such as inter-feature relationships in the learned representations, which downstream predictors can exploit effectively.
> GraphAny, in contrast, relies on a pseudo-inverse–based linear predictor to capture this information, which is less expressive and brittle to noise and feature-type.
>
> Finally, a *node–feature dynamic aggregation* mechanism allows GVT to adaptively determine how each node–feature is aggregated at every layer. In contrast, GraphAny relies on a static, predictor-imposed aggregation.
> Thereby, GVT exhibits substantially higher expressivity in capturing structural patterns across diverse graphs.
>
> ---
>
> We hope the above responses and the revised manuscript sufficiently address your concerns. If you have any further questions or suggestions, we would be glad to discuss them. Thank you for the considerable time and care you put into evaluating our work.

---

> > ### Comment · Reviewer_M8Kc · 2025-11-27
> >
> > Thanks for the response. Although overall I still feel the contribution and novelty are somewhat borderline, the newly added appendix shows the authors’ work and efforts, and I have raised my score accordingly.

---

> > > ### Author Response · Authors · 2025-11-28
> > >
> > > Thank you again for your thoughtful review. We sincerely appreciate your time and consideration.

---

### Official Review · Reviewer_vCdh · 2025-10-29

**Soundness:** 3
**Presentation:** 4
**Contribution:** 3
**Rating:** 6
**Confidence:** 4

**Summary:**

This paper tackles cross-dataset generalization for graph learning by introducing a “view space” that can encode graphs of any size and feature specification, enabling parameter sharing across incompatible node feature spaces. The authors propose Graph View Transformation (GVT), a learned mapping that projects any node-feature matrix through this shared space, and build Recurrent GVT (RGVT) on top as a foundation model for universally shareable graph knowledge. Trained on OGBN-Arxiv and tested on 27 node classification benchmarks, RGVT outperforms GraphAny and beats a dozen tuned GNN baselines. The results suggest that learning in the view space provides a principled and effective route to general node classifier, with code and data released for reproducibility.

**Strengths:**

1. The heterogeneous feature space is an important problem for GFM. The proposed view-space formulation is a compact and novel solution.
2. Theoreticall study on properties and expressivity results help justify the design.
3. Extensive experiments show consistent improvements with statistical significance
4. Writing is concise, figures are readable, and the narrative is easy to follow.

**Weaknesses:**

1. The statement given by the title is too strong. The proposed method provides a general model for node classification. Edge level and graph level tasks are not explored, and coverage of graphs without node features or with edge features is not addressed.
2. The choices for view finders is unclear. The paper does not clearly state how many views are used, how each view is constructed, or how sensitive performance is to these choices.
3. There is no complexity analysis. It would be better to provide a comparison of the the training and inference cost with GraphAny and other baselines.

**Questions:**

1. Is there a theoretical or empirical guideline for selecting the number of views?
2. Can the framework extend to graphs without node features, to graphs with edge features, and to other downstream tasks such as link prediction and graph classification? If so, what changes are required in the view transformation or the predictor?

---

> ### Author Response · Authors · 2025-11-22
>
> We sincerely appreciate your careful evaluation and insightful feedback.
> We first refer you to the common response above, where we clarify the sources of the perceived overstatement and summarize the main revisions made in the manuscript to address these concerns.
>
> ---
>
> **W1. Overstated Title, Mismatch Between the Positioning of the Work and Evaluation**
>
> As clarified in the common response, our work focuses on fully inductive node representation learning (FI-NRL), with node classification used to evaluate the learned representations.
> We thank you for the helpful feedback, the clarifications are now reflected in the revised title, *“Fully Inductive Node Representation Learning via Graph View Transformation.”*
> We hope that the revised title and manuscript address your concerns.
>
> ---
>
> **W2, Q1. The choices for view finders is unclear. The paper does not clearly state how many views are used, how each view is constructed, or how sensitive performance is to these choices.
> Is there a theoretical or empirical guideline for selecting the number of views?**
>
> Thank you for the detailed feedback.
> We clarify the candidate set of view finders in Section 8 with an explicit formulation (lines 395–401).
> We also include the selected configurations obtained through hyperparameter search, which are shared for all datasets, in Appendix J (lines 1418–1421).
>
> To further address your concerns, we conduct additional experiments examining how different view finders affect RGVT’s performance and provide practical guidelines (Appendix I).
> Our results show that RGVT remains stable as long as diffusion-style filters (symmetrically normalized or random-walk) are used.
> In contrast, Chebyshev filters degrade performance as the effective hop size increases due to amplified high-frequency noise, and any filter set that includes Chebyshev shows similar degradation.
> Thereby, we recommend using a mixture of diffusion-style graph convolution filters (symmetric normalization or random walk) with a moderate number of hops (2-3).
>
> ---
>
> **W3. There is no complexity analysis. It would be better to provide a comparison of the training and inference costs of GraphAny and other baselines.**
>
> In response, we conduct both a theoretical analysis and empirical runtime measurements of RGVT, along with baseline comparisons (Appendix H).
> Theoretically, RGVT scales linearly with graph size and feature dimensionality.
> Empirically, RGVT takes about 10 minutes to pre-train on OGBN-Arxiv and performs inference across all 28 datasets in 0.05 seconds.
> While these numbers are slower than simple GNNs, the absolute runtime remains reasonable, and the overall inference time is much faster than GraphAny, which takes 0.57 seconds.
>
> We further note that RGVT’s fully inductive capability eliminates the need for per-dataset hyperparameter search when adapting to new datasets, whereas GNNs require such tuning for strong performance.
> Additionally, RGVT’s representations can be reused even when the label space changes, while GraphAny must recompute its linear predictors in such cases.
> Taken together, these demonstrate that RGVT is a reliable yet practically efficient method.
>
> ---
>
> **W2, Q2. Can the framework extend to graphs without node features, to graphs with edge features, and to other downstream tasks such as link prediction and graph classification? If so, what changes are required in the view transformation or the predictor?**
>
> As clarified in the common response and the revised manuscript, this work focuses on fully inductive node representation learning (FI-NRL).
> A natural direction for future research is extending the framework toward more general-purpose representations that support diverse tasks, as well as adapting it to graphs with edge features or higher-order structures such as hypergraphs.
> We included discussion of limitations and future works of this work in the conclusion section (lines 518–525).
>
> For graphs without node features, the three airport-traffic datasets are included using one-hot features in the main experiments.
> RGVT consistently outperforms GraphAny on all three datasets in average +25.8\% and achieves at least the second-best performance among the 12 dataset-specialized GNNs.
>
> We additionally evaluate alternative feature-construction strategies to identify more reliable option for RGVT with non-attributed graphs (Appendix F).
> In summary, RGVT paired with DeepWalk embeddings yields strong and consistent performance, outperforming the dataset-specialized GCN on 5 out of 6 graphs.
>
> ---
>
> We hope the above responses and the revised manuscript sufficiently address your concerns. If any questions or concerns remain, please feel free to let us know. Thank you again for the time and care you invested in evaluating our work.

---

### Official Review · Reviewer_85dN · 2025-10-31

**Soundness:** 2
**Presentation:** 3
**Contribution:** 2
**Rating:** 2
**Confidence:** 4

**Summary:**

This paper tackles a critical problem for graph foundation models: datasets often have incompatible features. The authors solve this by proposing the view space, a novel representation axis that is independent of node features. They then introduce Graph View Transformation (GVT), a theoretically universal function that learns representations within this new space. GVT serves as the core building block for their RGVT foundation model. Impressively, experiments show the pre-trained, frozen RGVT model surpassed 12 specialized GNNs that were individually tuned for their specific tasks.

**Strengths:**

1. Novelty: The paper introduces the "view space," a new representation paradigm for graphs. This is a significant conceptual leap, as it bypasses the central problem of feature heterogeneity (incompatible dimensions and semantics) rather than trying to fix it with alignment or projection, which is what most prior work does.
2. Theoretical Grounding: The authors provide a solid theoretical foundation for their method. They mathematically prove that the proposed Graph View Transformation (GVT) achieves "dual permutation equivariance" (for both nodes and features), which formally guarantees its universality across graphs with arbitrary feature specifications.
3. Strong Results: The experiments provide compelling evidence for the "foundation model" claim. A single, pre-trained, and frozen RGVT model, with only a lightweight predictor, was able to outperform 12 different, specialized GNN models that were fully and individually tuned for each of the 27 downstream tasks. This demonstrates a remarkable level of generalizable knowledge transfer.

**Weaknesses:**

1. Overclaim. This manuscript claims that the proposed model is the first graph foundation model that is unlimited by the feature space barrier, enabling universal knowledge transfer. However, recent advances in text-free multi-domain graph pre-training generally do not struggle with the feature space heterogeneity, supporting the knowledge transfer across different graphs. Also, SAMGPT (WWW25) can be viewed as an initial success of GFM in this direction.
2. Miss of related work. The authors state that `` GraphAny, the only universal graph model to date". However, there has been a series of GFM attempts very recently.
3. Weak evaluations. RGVT is validated in the node classification task. However, as a general-purpose foundation model, important tasks, such as link prediction, graph classification, and node clustering, are touched on in the experiment.

**Questions:**

1. How does the model scale for graphs with no features?
2. GVT processes each feature channel independently. How does the model learn interactions between different features if they never mix?
3. By treating all features identically (to ensure feature equivariance), doesn't the model lose the ability to apply feature-specific logic, such as processing categorical and numerical features differently?
4. How sensitive is performance to the specific set of view finders?
5. How does the model's computational cost scale with a large number of features? A runtime benchmark against standard GNNs (which project the feature dimension to a small dimension) is missing.

---

> ### Author Response · Authors · 2025-11-22
>
> We sincerely appreciate your thorough review and insightful comments. Prior to the point-by-point response, please see the common response above, where we clarify the cause of the impression of overstatement and summarize the major revisions introduced to resolve it.
>
> ---
>
> **W1. Overclaim.**
>
> As noted in the common response, we no longer present our work as a graph foundation model; in the revised manuscript, we clearly position it within the fully inductive setting.
> GFMs that aim to generalize across different feature spaces, such as SAMGPT (WWW’25), still require per-dataset fine-tuning; therefore, they cannot directly operate on unseen feature spaces in a fully inductive manner.
> To clarify this distinction, Section 7 (Related Works) now explicitly explains the difference between GFMs' feature generalization (including SAMGPT) and the fully inductive setting (lines 352–356).
>
> ---
>
> **W2. Miss at Related Work.**
>
> Thank you for pointing this out.
> As in the common response, we no longer use the term “universal”; we now refer to them precisely as fully inductive.
> Regarding prior work on fully inductive graph learning, we acknowledge that recent studies [1,2] also pursue fully inductive node classification.
> Accordingly, we removed the phrase “up to date” when introducing GraphAny throughout the manuscript.
> Because the works [1,2] are currently submissions, GraphAny remains the most practically recognized state-of-the-art approach in the fully inductive setting.
>
> We further note that, although several fully inductive approaches have been proposed, they all make a classification prediction directly from the input features; our work focuses on a different problem: *representation learning*, which aims to provide a reusable node representation for arbitrary graphs.
>
> [1] Bringing Graphs to the Table: Zero-Shot Node Classification via Tabular Foundation Models
> (ICLR 2026 submission)\
> [2] GraphPFN: A Prior-Data Fitted Graph Foundation Model (ICLR 2026 submission)
>
> ---
>
> **W3. Weak Evaluation as a General-purpose Foundation Model.**
>
> As clarified, the scope of this work is node representation learning with node classification as the primary use case, which we now state clearly throughout the revised paper, including in the limitations paragraph (lines 518–525).
> Extending the framework toward general-purpose representation learning that can support a broader range of tasks is an important direction for future work.
>
> ---
>
> **Q1. How does the model scale for graphs with no features?**
>
> In the main experiment, the three airport-traffic datasets without node attributes are handled using one-hot features (following GraphAny's experimental setting).
> RGVT consistently outperforms GraphAny on all three datasets with an average increase of +25.8\%, and it achieves at least the second-best performance among the 12 dataset-specialized GNNs.
>
> In response to your concern about the scalability of one-hot encoding, we conduct additional experiments using more realistic feature-construction methods (Appendix F).
> The results show that RGVT paired with DeepWalk embeddings performs strongly and consistently on non-attributed graphs, outperforming dataset-specialized GCN on 5 out of 6 graphs.

---

> ### Author Response · Authors · 2025-11-22
>
> **Q2. How does GVT capture cross-feature interactions if it processes each feature channel independently?**
>
> Thank you for the thoughtful question.
> GVT itself does not model cross-feature interactions, as each feature channel is processed independently by design.
> However, the feature-permutation-equivariant condition of GVT (Section 3) allows that cross-feature structure is preserved in the resulting representations, unlike permutation-invariant designs that discard such information.
> As a result, the per-dataset predictor---operating on the produced node representations---can freely combine information across feature channels and capture these interactions.
> We have clarified this point in the revised manuscript with theoretical proofs (lines 143–149).
>
> ---
>
> **Q3. By treating all features identically, doesn't the model lose the ability to apply feature-specific logic, such as processing categorical and numerical features differently?**
>
> This is an important aspect of fully inductive learning.
> In a fully inductive setting, the model must operate on *any* type of feature without knowing in advance what kinds of features it will encounter.
> If the model learns feature-specific rules, such as “processing categorical features differently from numerical ones,” those rules will break as soon as it encounters graphs with LLM-embedded features or any other unfamiliar feature format.
> Thus, treating features uniformly is essential for maintaining full inductiveness.
> The strong performance of RGVT, which is pre-trained on word embeddings (of the OGBN-Arxiv dataset) but is successfully applied to datasets with one-hot, categorical, and numerical features, further supports this principle.
> Our additional experiments show that RGVT also performs well on test graphs with LLM-embedded features (Appendix G).
>
> At the same time, treating features identically does not mean losing feature distinctions.
> As in the response to the previous question, GVT’s feature-permutation-equivariant design preserves feature-wise information, allowing the downstream predictor to exploit it as needed.
>
> ---
>
> **Q4. How sensitive is performance to the specific set of view finders?**
>
> To address this concern, we conduct experiments evaluating various view-finder configurations, including random-walk, symmetrically normalized, and Chebyshev polynomial filters.
> We vary hop sizes, filter combinations, and summarize the resulting selection guidelines (Appendix I).
>
> Our findings show that RGVT’s performance remains stable as long as diffusion-style filters (symmetric normalized or random-walk) are used.
> In particular, symmetric normalized filters alone provide consistently strong results over the best-performing GNNs across all hop sizes.
> In contrast, Chebyshev filters degrade performance as the effective hop size increases due to amplified high-frequency noise, and any filter set that includes Chebyshev suffers from the same issue.
>
> ---
>
> **Q5. How does the model's computational cost scale with a large number of features?**
>
> In response, we provide theoretical and empirical analyses of RGVT’s computational cost (Appendix H).
> Theoretically, RGVT scales linearly in both graph size and feature dimensionality.
> For datasets with large feature dimensions, the main computational bottleneck is the dense matrix multiplication in the dimension-collapsing mapping of GVT, which has a complexity of O(NFC^2).
> Here, C denotes the number of view finders used in RGVT.
> In comparison, the matrix multiplication in standard GNN layers has a complexity of O(NH^2), where H is the hidden dimension.
> Under our configuration (H = 128, C = 5), this results in a relative factor of F/655.
> In the dataset with largest feature in our 28 benchmark suite (F = 8701), this translates to at most a 13.3x slowdown in the dense transformation.
>
> Empirically, however, RGVT remains practical: training takes about 10 minutes on OGBN-Arxiv, and inference across all 28 datasets takes 0.05 seconds, which is slower than typical GNNs but still reasonable.
> Crucially, RGVT is *fully inductive* and can be applied directly to new datasets, whereas GNNs require substantial per-dataset training and search.
> Moreover, as a *representation learning* framework, RGVT produces reusable node embeddings that remain valid even when the label space changes.
> These properties offer meaningful computational advantages in real-world applications.
>
> ---
>
> We hope the above responses and the revised manuscript sufficiently address your concerns.
> Please let us know if you have any further questions or comments, we sincerely appreciate the time and care you invested in this review.

---

### Author Response · Authors · 2025-11-22
**Summary of Manuscript-wide Revisions**

We sincerely appreciate the reviewers’ thoughtful comments.
After carefully considering the feedback, we realized that our terminology and presentation---especially the terms “universality” and “graph foundation models,” for which we believed our method was designed---unintentionally created a shared impression of overstatement.
We apologize for this misunderstanding and provide the following clarifications and revisions as a common response to all reviewers.

**Introduction to FI-NRL.**
Using more precise terminology, our goal is to solve *fully inductive node representation learning* (FI-NRL): learning a single model that can produce node representations for arbitrary graphs without any additional training.
This capability is important, paralleling the success of pretrained encoders in NLP and CV, where one encoder provides reusable representations across datasets and only a lightweight predictor is specialized for each dataset and task.

FI-NRL is particularly challenging because graph datasets often have incompatible feature spaces, requiring a model to operate on feature spaces it has never seen.
GraphAny achieves fully inductive inference by attending over candidate predictions.
This idea is novel, but GraphAny is not an encoder and cannot produce reusable representations.
As a result, the problem has remained unsolved. Our view-space framework aims to tackle this problem for the first time in the literature.

**Origins of the Misguidance.**
The misunderstanding primarily arose from our overemphasis on fully inductive learning.
(1) We used the term “universal” interchangeably with “fully inductive” to highlight that the model can process arbitrary graphs. This was inaccurate terminology and may have implied stronger claims than intended.
(2) We also presented full inductiveness as the defining property of a graph foundation model, which was oversimplification; we acknowledge that task-level and structural generalization are equally (even more) important characteristics of a foundation model.
As a result, our use of the term “graph foundation model” unintentionally overstated the level of generalization and mischaracterized RGVT and GraphAny as universal GFMs.
We sincerely apologize for these errors and the resulting confusion.

**Overall Revision Direction.**
To address these issues, we have revised the manuscript to use the precise term “fully inductive node representation learning” and clarified its meaning in more detail.
We no longer use the term “universal,” nor do we refer to RGVT (Ours) or GraphAny as a “GFM.”
The title, abstract, introduction, and main text have been updated to more accurately reflect our intended goals and contributions.
All revised passages are highlighted in blue in the updated manuscript.
We hope these updates fully address the reviewers’ concerns about the unintended overstatements and clearly clarify our intention.

The modifications to the original submission are summarized as follows:
- **(Title and Abstract)** Revised to more clearly reflect the goal and scope of the work.
- **(Lines 44--58)** Revised the introduction to motivate inductive learning on graphs especially, with a focus on handling unseen feature spaces.
- **(Lines 59--67, 110--118)** Added a clear motivation and formulation of fully inductive node representation learning.
- **(Lines 347--357)** Updated the related-work section to clearly articulate the connection to GFM approaches and the differences in problem setting.
- **(Lines 518--525)** In the conclusion, explicitly clarified the scope, limitations, and future directions of the present work.

The experimental results newly included to the paper are as follows:
- **(Appendix F)** Evaluation on **non-attributed graphs** using several feature-construction strategies.
- **(Appendix G)** Evaluation on **graphs with LLM-embedded features**.
- **(Appendix H)** Theoretical and empirical **complexity analyais** covering training, adaptation, and inference costs.
- **(Appendix I)** Experiments demonstrating how view finder affect performance, along with recommended **guidelines for selecting view finder sets**.

---

### Author Response · Authors · 2025-11-26

Dear Reviewers,

Thank you for taking the time to evaluate our work. We have carefully considered your comments and incorporated the corresponding revisions and additional experiments into the manuscript, along with detailed responses. We would appreciate your guidance on whether these updates sufficiently address your concerns. If any issues remain, we would glad to make further revisions or conduct additional experiments. Your feedback is highly valuable in strengthening our work.

Regards,

The Authors

---

### Author Response · Authors · 2025-12-02
**Rebuttal Summary for the Area Chairs**

Dear Area Chairs,

We sincerely thank the ACs for facilitating the review process, especially given the additional challenges created by the identity-leak issue.
We deeply appreciate your hard work, which plays a critical role in maintaining the fairness and strength of our community.
Below we present a summary of the reviewers’ concerns and the improvements we made during the rebuttal to address them.

- **Impression of over-claiming** (85dN, vCdh, M8Kc):
In the initial submission, we used the term “universal” to indicate that the model can operate on arbitrary unseen graphs and assumed this as a defining property of a “graph foundation model (GFM).”
This unintentionally created an impression of over-claiming.
To address this, we removed the use of these terms and clarified the precise scope and intent of the work throughout the manuscript.
We now explicitly define our goal as *fully inductive node representation learning* (FI-NRL), in which a model can produce node representations for arbitrary unseen graphs.
Further details on corresponding revisions are provided in the prior common response.

- **Lack of experiments on view-finder selections** (85dN, vCdh):
We added experiments on various view-finder configurations, including different filter types, hop sizes, and their combinations along with the selection guidelines (Appendix I).

- **Missing complexity analysis** (85dN, vCdh):
We added both theoretical and empirical complexity analyses, compared against baselines, and clarified the practical time cost and complexity advantages of the framework (Appendix H).

- **Handling non-attributed graphs** (85dN, vCdh):
Beyond the one-hot features used for non-attributed datasets in the main experiments, we evaluated additional feature-construction strategies. Based on these results, we recommend the DeepWalk + RGVT combination (Appendix F).

- **Handling LLM-embedded feature graphs** (M8Kc):
We added experiments on graphs with LLM-embedded features and showed that RGVT remains effective (Appendix G).

- **Concerns regarding feature independence** (85dN):
We clarified that GVT’s *feature-permutation-equivariance* allows the lightweight predictor to capture cross-feature interactions.

- **Concerns regarding heterophilous graphs** (M8Kc):
We clarified that GVT’s *node-feature dynamic aggregation* effectively handles the complex patterns of heterophilous graphs which is further supported by strong results across 11 heterophilous datasets in the main experiments.

While one reviewer (M8Kc) increased their score from 4 to 6, we were unable to receive follow-up comments from the other two reviewers (85dN, vCdh) due to the early termination of the rebuttal phase.
We believe that the revisions, clarifications, and additional experiments included in our response successfully addressed their concerns.

We also deeply appreciate the reviewers’ recognition of our work’s strengths, including its **novelty** (85dN, vCdh), **theoretical grounding** (85dN, vCdh), **strong empirical performance** (85dN, vCdh, M8Kc), and **presentation quality** (85dN, vCdh, M8Kc).
These remarks reinforce our confidence in the contribution and potential impact of our approach. In addition to the method itself, we believe our work provides two unique insights for the community:

- The proposed view space provides a standardized way to represent graphs, analogous to pixels for images or characters for text, establishing a foundation for building fully inductive graph learning methods.

- The strong performance of graph view transformation invites a fresh look at a long-standing assumption in graph learning: feature transformation does not have to be the default path of capturing graph structure.

We would be grateful for the opportunity to share this work and its insights with the community. Thank you again for your time and support.

Best regards,

Authors

---

### Meta-Review · Area_Chair_Xqi7 · 2026-01-13

**Summary:**

The reviewers acknowledged the novelty and technical strength of the proposed view space formulation and the strong empirical performance of the RGVT model, particularly its ability to outperform GraphAny and multiple tuned GNN baselines on node classification benchmarks. Reviewers highlighted the conceptual contribution of introducing a representation axis that bypasses feature heterogeneity, as well as solid theoretical analysis establishing node- and feature-permutation equivariance (e.g., Reviewer 85dN: “a significant conceptual leap” and “solid theoretical foundation”; Reviewer vCdh: “compact and novel solution”).

**However, the reviewers raised substantial concerns that ultimately outweigh these strengths.** The most critical issues include:
- Overstated positioning and scope, especially the use of terms such as “universal” and “graph foundation model,” which reviewers consistently argued were not supported by the evaluation scope (node classification only) (Reviewers 85dN, vCdh, M8Kc).
- Limited task coverage, with no evaluation on link prediction, graph-level tasks, or other settings expected of a general or foundational representation learner (Reviewers 85dN, vCdh, M8Kc).
- Incomplete justification of key design choices, including how cross-feature interactions are captured when feature channels are processed independently, and whether the recurrent, weight-shared architecture has sufficient expressivity, especially on heterophilous graphs (Reviewers 85dN, M8Kc).
- Concerns about relevance and positioning relative to recent related work, particularly other fully inductive or foundation-style graph models, which were initially underrepresented or overstated in the submission (Reviewers 85dN, M8Kc).

While the rebuttal addressed several presentation and clarity issues, the reviewers’ core concerns about scope, task generality, and the strength of the claims relative to evidence remain only partially resolved, motivating a rejection recommendation.

**Reviewer Concerns:**

## Concerns discussed/addressed (partially) in the rebuttal

- **Over-claiming and terminology (“universal”, “graph foundation model”)**
  - Multiple reviewers criticized the original framing as overstated (Reviewer 85dN: “claims … universal knowledge transfer”; Reviewer vCdh: “statement given by the title is too strong”; Reviewer M8Kc: “you can't say it's universal”).
  - The authors explicitly acknowledged this issue, removed the term “universal,” and repositioned the work as fully inductive node representation learning (FI-NRL), revising the title, abstract, and conclusion accordingly. This directly addresses the reviewers’ criticism at the level of terminology and stated scope.

- **Missing complexity analysis and runtime discussion**
  - Reviewer 85dN and vCdh both pointed out the lack of complexity analysis and runtime benchmarks (“A runtime benchmark … is missing”; “There is no complexity analysis”).
  - The rebuttal added both theoretical and empirical complexity analyses (Appendix H), including scaling with feature dimensionality and comparisons to GraphAny. This concern is reasonably addressed.

- **Unclear choice and sensitivity of view finders**
  - Reviewer vCdh explicitly noted that “the choices for view finders is unclear.”
  - The authors added ablation studies and guidelines on view-finder selection (Appendix I), clarifying stability with diffusion-style filters. This concern was substantively addressed.

## Concerns that remain outstanding:

- **Limited task scope relative to claims of generality**
  - All three reviewers emphasized that the evaluation is restricted to node classification and does not cover link prediction, graph-level tasks, or other downstream uses expected of a broadly reusable representation model (Reviewer 85dN: “important tasks … are touched on”; Reviewer vCdh; Reviewer M8Kc).
  - While the authors clarified scope and added limitations, no new evidence was provided to demonstrate broader applicability. This remains a fundamental limitation.

- **Model expressivity and cross-feature interaction**
  - Reviewer 85dN questioned how cross-feature interactions are learned when “GVT processes each feature channel independently,” and Reviewer M8Kc raised concerns about expressivity under recurrent weight sharing, especially for heterophilous graphs.
  - The rebuttal provided conceptual explanations (delegating interactions to the predictor; dynamic aggregation) but did not provide direct empirical validation or ablations to demonstrate that these design choices are sufficient. These concerns remain partially unresolved.

- **Broader positioning relative to recent related work and alternative paradigms**
  - Reviewers (especially 85dN and M8Kc) argued that feature heterogeneity may not be the primary bottleneck for graph foundation models and that recent models (e.g., tabular or type-aware approaches) weaken the novelty claim.
  - Although the authors clarified their perspective and repositioned the work, this remains a conceptual disagreement that the rebuttal does not conclusively resolve.

**Reviewer Scores:**

- **Reviewer 85dN (Score: 2)**: This reviewer acknowledged strong novelty and empirical performance but raised persistent concerns about overclaiming, limited task scope, and missing evaluations beyond node classification. Although the rebuttal corrected terminology and added analyses, the reviewer’s core objections regarding scope and generality would likely remain.
  - Expected score change: No change (likely remains at 2).

- **Reviewer vCdh (Score: 6)**: This reviewer was generally positive but explicitly conditioned acceptance on clarifying scope, view-finder choices, and complexity analysis. These points were largely addressed in the rebuttal, which likely strengthens their confidence in the technical contribution. However, given the continued limitation to node classification, it is unlikely this reviewer would significantly increase their score.
  - Expected score change: Slight increase or unchanged (likely 6 → 8 or 6).

- **Reviewer M8Kc (Score: initially 4, later increased)**:
This reviewer was the most critical of the foundation-model framing and conceptual motivation but explicitly stated after the rebuttal that “the newly added appendix shows the authors’ work and efforts, and I have raised my score accordingly.”
  - Expected score change: Moderate increase (approximately 4 → 6), but still with reservations about novelty and scope.

---

### Decision · Program_Chairs · 2026-01-26

Reject